# AI Engram: In Search of Memory Traces in Artificial Intelligence

**Jea Kwon** [* 1]   **Dong-Kyum Kim** [* 1]   **Jiwon Kim** [* 1 2]   **Yonghyun Kim** [1]   **Woong Kook** [2]   **Meeyoung Cha** [1 3]

## Abstract

Memory formation is fundamental to intelligence, yet whether deep neural networks preserve identifiable memory traces analogous to biological memory units remains an open question. This work introduces a geometric framework to identify such "AI engrams" by formalizing the neuroscientific criteria of *specificity, reactivation, sufficiency*, and *necessity* into a constrained inverse problem. We derive a closed-form estimator that isolates individual memory traces from globally entangled parameters, and show that this biologically-derived solution corresponds to a natural gradient update on the parameter manifold. AI engrams enable surgical manipulation of learned knowledge: any subset of memories can be composed or erased through linear arithmetic, without iterative optimization. Experiments ranging from simple MLPs to LLMs demonstrate the causal validity and substantial scalability of AI engrams. Together, these results bridge theories of biological memory and artificial representation learning and offer geometric insight into how deep networks simultaneously support functional specificity within distributed storage.

## 1. Introduction

Memory is the foundation of intelligence (Hawkins & Blakeslee, 2004). Intelligence emerges when systems learn to construct meaningful representation of the world (Bengio et al., 2013), suggesting that memory functions not as passive storage but as the structural representations that model reality (Ha & Schmidhuber, 2018). This work begins from the premise that representations form as concrete, identifiable structures within a neural network, analogous to **neural engrams** (Semon, 1921). Building on this idea, we

---
[*]Equal contribution [1]Max Planck Institute for Security and Privacy (MPI-SP), Bochum, Germany [2]Seoul National University (SNU), Seoul, Korea [3]KAIST, Daejeon, Korea. Correspondence to: Meeyoung Cha <mia.cha@mpi-sp.org>.

*Proceedings of the 43rd International Conference on Machine Learning*, Seoul, South Korea. PMLR 306, 2026. Copyright 2026 by the author(s).

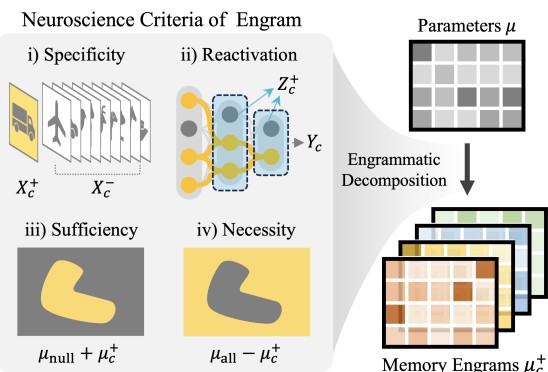

*Figure 1.* **Illustration of neuroscience criteria for AI engram.** The engram $\boldsymbol{\mu}_c^+$ of target concept $c$ is identified as the causal substrate satisfying: i) **Specificity**: selective encoding of target experience $\boldsymbol{X}_c^+$ relative to reference $\boldsymbol{X}_c^-$; ii) **Reactivation**: consistent reproduction of learned internal representations $\boldsymbol{Z}_c^+$; iii) **Sufficiency**: memory induction via injection of $\boldsymbol{\mu}_c^+$ into the null state $\boldsymbol{\mu}_{\text{null}}$; iv) **Necessity**: memory elimination via surgical ablation of $\boldsymbol{\mu}_c^+$ from the learned state $\boldsymbol{\mu}_{\text{all}}$.

introduce a method to locate such structures in a model's parameter–an AI parallel to the longstanding neuroscientific quest to pinpoint the physical substrate of memory.

Identifying physical memory traces in deep neural networks has long been intractable because learned parameters participate in many behaviors simultaneously, producing representations that are inherently distributed and entangled (Hinton, 1984; Elhage et al., 2022; Bau et al., 2017). This work introduces an analytical framework that isolates **AI engrams** from globally entangled parameters. Unlike heuristic attribution methods, we define neuroscience-based constraints that yield a closed-form solution and precisely recover memory traces directly from trained weights and activation statistics.

Specifically, we formulate the AI engram as an inverse problem governed by four canonical criteria (Fig. 1): *specificity, reactivation, sufficiency,* and *necessity* (Josselyn & Tonegawa, 2020; Tomé et al., 2024). By enforcing these constraints, we extract the unique optimal substrate responsible for a specific memory. Theoretical analysis shows that our framework allows an information-geometric interpretation, revealing its connection to the underlying curvature of the parameter manifold. By translating the biological criteria into algebraic optimization constraints, we show that these constraints naturally yield the minimum-norm

projection under the Fisher metric. The main contributions are summarized below, and the code is available at `https://github.com/jeakwon/ai-engram`.

- **A framework for identifying memory traces in deep neural networks**: By translating biological criteria into algebraic constraints, we derive a scalable closed-form estimator that isolates AI engrams.

- **Amortized memory isolation**: Our framework decouples memory traces from the entangled parameter manifold, enabling the instantaneous composition or erasure of knowledge via simple linear arithmetic. This approach resolves the combinatorial complexity of unlearning, demonstrating robustness across architectures from simple MLP to billion-parameter LLMs.

- **A link between engrams and information geometry**: We discover that our closed-form estimator from neuroscience-inspired constraints; it recovers the Fisher curvature, coinciding with the natural gradient direction on the parameter manifold.

## 2. Related Work

**Selective knowledge manipulation** Controlling the behavior of pre-trained models often relies on identifying and modifying the parameters that encode specific knowledge. In machine unlearning, such targeted interventions, typically via selective fine-tuning, can effectively remove the influence of designated training data (Fan et al., 2024). Model editing follows a similar principle, with advances such as Unified Concept Editing (UCE) (Gandikota et al., 2024) that extends earlier approaches like TIME (Orgad et al., 2023), MEMIT (Meng et al., 2023), and ROME (Meng et al., 2022) to enable simultaneous erasure and debiasing of multiple concepts without iterative optimization. Although UCE achieves efficiency through a closed-form solution that avoids gradient calculation, it requires computing covariance matrices of hidden activations. Our proposed method likewise provides a closed-form update but differs fundamentally by incorporating neuroscience-inspired constraints to define the memory trace.

**Network decomposition** A central challenge in interpreting neural networks is to break them into components whose structure and interactions are interpretable. Most existing approaches decompose hidden activations, using sparse autoencoders to recover monosemantic dictionary elements (Huben et al., 2024), yet these methods do not yield a functional decomposition of the parameters themselves (Sharkey et al., 2025). In parameter space, work on weight disentanglement has aimed to isolate functional subnetworks using binary masks (Mallya & Lazebnik, 2018;

Ramanujan et al., 2020; Panigrahi et al., 2023). Moving beyond masks, Task Arithmetic (Ilharco et al., 2023) showed that distinct capabilities can be linearly separated by treating model weights as vectors. However, these methods often operate at a coarse, global level.

Linear parameter decomposition (Braun et al., 2025) was introduced to address this issue by decomposing neural networks into subcomponents in parameter space. Braun et al. (2025) proposed Attribution-based Parameter Decomposition (APD) but it relies on gradient-based attribution and highly sensitive with the hyperparameter choice and not scalable. Stochastic Parameter Decomposition (SPD) (Bushnaq et al., 2025) has been proposed to address this issue by using learnable rank-one subcomponents. In contrast to previous works that yield non-unique subcomponents or rely on iterative gradient-based optimization, our method decomposes the model into unique subcomponents (or engrams) in a single forward pass.

## 3. Defining Learning and Memory

**Overview.** Modern neural networks store knowledge in a deeply entangled fashion, where a single weight matrix simultaneously supports many concepts with no obvious mapping from parameters to specific memories. This entanglement is the central obstacle to surgical knowledge manipulation, such as removing a concept without retraining, inserting a behavior without fine-tuning, or auditing what a model knows. The goal of this work is to identify the *causal substrate of a specific concept* within trained weights, the sub-component whose ablation removes the concept and whose injection imparts it, while leaving unrelated knowledge intact. Following classical neuroscience, we call this substrate an *engram* and formalize its identification by translating four canonical criteria from memory research (**specificity, reactivation, sufficiency, necessity**) into algebraic constraints on the parameter space.

We start by formally defining the learning of *experience D* (e.g., training data) for any learnable system.

**Definition 3.1** (**System and Experience**). We define a learnable system $f$ as the tuple $(\boldsymbol{\mu}, \boldsymbol{\pi})$, consisting of:

- **Mutable components ($\boldsymbol{\mu}$):** The plastic elements (e.g., synaptic weights) are updated through experience.

- **Immutable components ($\boldsymbol{\pi}$):** The fixed elements (e.g., architecture) that remain constant.

Given an input $\boldsymbol{X}$, the system produces an output $\boldsymbol{Y} = f(\boldsymbol{X}; \boldsymbol{\mu}, \boldsymbol{\pi})$. We denote by $\boldsymbol{\mu}_t$ the memory state at time $t$, with $t = 0$ and $t = 1$ corresponding to the states before and after the learning of experience $D$.

**Definition 3.2** (**Concept**). We define a *concept $\mathcal{C}$* as a functional specification governing how the mutable parameter $\boldsymbol{\mu}$ adapts under architecture $\boldsymbol{\pi}$ by partitioning the input space:

- **Target region ($X^+$):** Inputs requiring **gain-of-function** to encode new representations.

- **Reference region ($X^-$):** Inputs for which existing behaviour must exhibit **conservation-of-function** within the null space of prior representations.

### 3.1. Engram Identification as an Inverse Problem

In realistic learning scenarios, the total parameter update $\Delta\mu = \mu_1 - \mu_0$ is driven by mixed experiences $D = \{X^+, X^-\}$, where $X^+$ denotes the target inputs and $X^-$ represents prior or unrelated knowledge. Because learning must simultaneously acquire new behavior on $X^+$ while preserving existing behavior on $X^-$, the resulting update $\Delta\mu$ becomes **physically entangled**, with multiple memory traces superposed in overlapping parameter directions.

Isolating the unique engram $\mu^+$ is thus an **inverse problem** constrained by two biological principles (see Fig. 1). First, **representational nature** requires (i) *specificity* for target $X^+$ over reference inputs $X^-$ and (ii) selective *reactivation* of internal representation $Z_t$. Second, **causal validity** requires that $\mu^+$ satisfies (iii) *sufficiency* to induce learned behavior via induction and (iv) *necessity* for its expression via ablation.

Concrete instantiation. For the CIFAR-10 cat-unlearning scenario, the concept $\mathcal{C}$ = "cat" partitions the training data into a target set $X^+$ (cat images) and a reference set $X^-$ (the remaining nine classes). The four neuroscientific criteria translate concretely: *specificity* requires $\mu_{\text{cat}}^+$ to remain inert on non-cat images, *reactivation* requires it to reproduce the network's cat-specific internal states, *sufficiency* requires that injecting it into a naive model imparts cat-recognition, and *necessity* requires that ablating it removes cat-recognition while preserving the other nine classes. The remainder of the paper develops a closed-form estimator for $\mu_{\text{cat}}^+$ and its multi-concept generalization.

## 4. AI Engrams in Deep Neural Networks

### 4.1. Synaptic Decomposition of Memory

In a deep neural network, the mutable parameter set $\mu$ encompasses all trainable weights across every layer, expressed as $\mu = \{W^{(1)}, W^{(2)}, \ldots, W^{(L)}\}$. For a given experience $X^+$, we define the corresponding **memory engram $\mu^+$** as the collection of *causal* parameter updates that this experience induces throughout the network:

$$\mu^+ = \{W^{+(1)}, \ldots, W^{+(L)}\}. \tag{1}$$

To make the engram-identification problem tractable, we decompose it into a layer-wise sub-problems. For each layer $l$, the objective is to recover the **synaptic engram $W^{+(l)}$**, the layer-specific component of the causal memory trace.

| Target | Baseline ($t=0$) | Learned ($t=1$) |
|---|---|---|
| *1. Observed States (Natural Learning)* | | |
| Target ($X^+$) | $W_0 X^+ = Z_0^+$ | $W_1 X^+ = Z_1^+$ |
| Ref. ($X^-$) | $W_0 X^- = Z_0^-$ | $W_1 X^- = Z_1^-$ |
| *2. Intervened States (Causal Manipulation)* | | |
| Target ($X^+$) | $^\dagger (W_1 - W^+)X^+ \to \bar{Z}_0^+$ | $^\ddagger (W_0 + W^+)X^+ \to \bar{Z}_1^+$ |
| Ref. ($X^-$) | $(W_0 + W^+)X^- \to \bar{Z}_0^-$ | $(W_1 - W^+)X^- \to \bar{Z}_1^-$ |

*Table 1.* **Operationalizing Engrammatic Axioms as Optimization Constraints.** By contrasting observed states with intervened states, we establish formal linear-algebraic constraints from four neuroscience engram criteria: **i) Specificity** ($X^\pm$, target vs. ref.), **ii) Reactivation** ($\bar{Z}_{\{0,1\}}^\pm$, internal representation), **iii) Sufficiency**(‡, gain-of-function) and **iv) Necessity** (†, loss-of-function).

### 4.2. Retrospective Formulation in Linear $Z$-Space

We formalize the synaptic engram in the pre-activation space $Z = WX$, where the effect of weight updates can be isolated without non-linear distortions.[1] Because the activation function $\sigma(\cdot)$ is fixed, matching the internal state $Z$ provides a sufficient—and stricter—condition for reproducing the functional output $Y = \sigma(Z)$.

Engram identification is performed **retrospectively** on the fully trained network ($t=1$). Utilizing the input statistics $X$ at this final state grounds the analysis in the **stable representational manifold** where the target memory resides, ensuring that the isolated engram reflects the model's converged causal structure.

### 4.3. Algebraic Constraint Framework

We identify $W^+$ by contrasting the **observed states** produced during natural learning with the **intervened states** generated through algebraic manipulation. This contrast isolates the specific weight changes supporting the expression of the learned memory (as detailed by the two state conditions in Table 1).

The formal constraints specify that the synaptic engram $W^+$ isolates the component of $\Delta W$ that selectively reconstructs the target memory $X^+$ while leaving the reference subspace inert. To meet **sufficiency**, injecting the engram must reproduce the learned internal state (yielding, $\bar{Z}_1^+ \approx Z_1^+$), while ablating it must restore the naïve pre-learning state ($\bar{Z}_0^+ \approx Z_0^+$). In parallel, strict **specificity** demands that the engram remain inert for all reference experiences $X^-$, such that neither injection nor ablation perturbs their naturally observed states ($\bar{Z} \approx Z$).

---

[1] Layer superscripts are omitted for brevity; bias terms are absorbed into $W$ via input augmentation.

## 4.4. Deriving the Optimization Objective

We seek the optimal synaptic engram $W^+$ that minimizes the discrepancy between the intervened and observed states across all conditions. By construction, this optimal engram is the loss-minimizing solution that best satisfies sufficiency, necessity, and specificity within the trained manifold and can be defined as the sum of squared Frobenius norms:

$$\mathcal{L}(W^+) = \underbrace{\left\| \bar{Z}_1^+ - Z_1^+ \right\|_F^2 + \left\| \bar{Z}_0^+ - Z_0^+ \right\|_F^2}_{\text{Causal Reactivation}}$$
$$+ \underbrace{\left\| \bar{Z}_0^- - Z_0^- \right\|_F^2 + \left\| \bar{Z}_1^- - Z_1^- \right\|_F^2}_{\text{Target Specificity}}.$$

Substituting the intervened state definitions from Table 1, the objective collapses into a **canonical dual-form**:

$$\mathcal{L}(W^+) = 2\|(W^+ - \Delta W)X^+\|_F^2 + 2\|W^+ X^-\|_F^2. \quad (2)$$

Crucially, Eq. (2) demonstrates that engram identification is equivalent to isolating the component of the layer-wise update $\Delta W$ that accounts for the target memory while remaining orthogonal to the reference subspace. This provides a clean mathematical translation of the neuroscientific engram definition into a scalable optimization problem.

**Proposition 4.1** (**Spectral AI Engram Estimator**). *Under the hard constraint of specificity ($W^+ X^- = 0$), the optimal synaptic engram $W^+$ that minimizes the discrepancy in Eq.* (2) *is given by the minimum-norm closed-form:*

$$W^+ = \Delta W \, \Sigma^+ \left( \Sigma^+ + \Sigma^- \right)^\dagger \quad (3)$$

*where $\Sigma^+ = X^+ X^{+\top}$ and $\Sigma^- = X^- X^{-\top}$ are uncentered covariance matrices.*

*Proof Sketch.* We introduce a Lagrange-multiplier matrix to enforce the null-space constraint and apply the Karush–Kuhn–Tucker (KKT) conditions. By exploiting the property of the Moore–Penrose pseudoinverse $\mathcal{X}^\dagger = \mathcal{X}^\top (\mathcal{X} \mathcal{X}^\top)^\dagger$, we decouple the memory footprint from the dataset size $N$ and reduce space complexity from linear $\mathcal{O}(Nd)$ to a constant $\mathcal{O}(d^2)$. Such scalability is essential for analyzing modern large-scale models where $N \gg d$. While the constraint $W^+ X^- = 0$ is imposed strictly in the Lagrangian, the resulting operator $P^+ = \Sigma^+ (\Sigma^+ + \Sigma^-)^\dagger$ acts as a soft, spectrally-weighted filter rather than a hard projector; we analyze its spectral structure in Section 6.3. A full derivation is provided in Appendix A.

Remarkably, our analytical estimator relies solely on local input statistics, effectively decoupling engram extraction from the entire output hierarchy $(Y, Z)$. This formulation eliminates the need for backpropagation or iterative optimization, recasting memory identification from a stochastic search into a deterministic, one-shot spectral estimation.

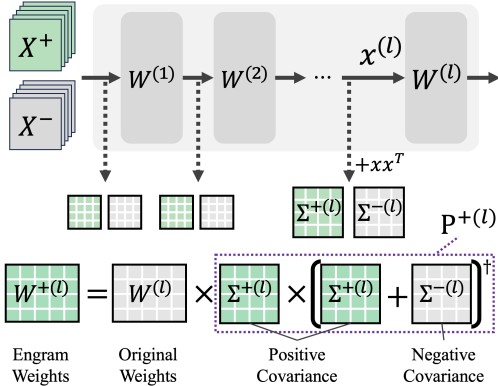

*Figure 2.* **Mechanics of the Engram Method.** We compute the Engram weights as $W^{+(l)} = W^{(l)} P^{+(l)}$, where the projection matrix $P^{+(l)} = \Sigma^{+(l)} (\Sigma^{+(l)} + \Sigma^{-(l)})^\dagger$ acts as a surgical filter, where covariances are accumulated during a single forward pass.

## 4.5. Practical Instantiation: Retrospective Estimation

Although the total update is defined as $\Delta W = W_1 - W_0$, random initialization $W_0$ typically acts as a high-entropy baseline. Under the *tabula rasa* (blank slate) assumption (Turing, 1950), this initial state contributes negligible structural information to the learned features. We therefore instantiate the update retrospectively as $\Delta W \approx W - 0$, where $W$ represents the fully trained weights. This leads to the final estimator in our experiments from Eq.(3) (See Fig. 2):

$$\boxed{W^+ = W \Sigma^+ (\Sigma^+ + \Sigma^-)^\dagger} \quad (4)$$

By grounding the estimation in the converged weights $W$, we isolate the engram directly from the stable representational manifold achieved after learning. Beyond simplification, this instantiation has a structural consequence: it decouples the layer-wise sub-problems, enabling Eq. (4) to be solved in a single forward pass across all layers in parallel; we formalize this property and contrast it with the delta-weight instantiation $\Delta W = W_{\text{ft}} - W_{\text{pt}}$ in Appendix G.

Ultimately, the total **AI engram** $\mu^+$ associated with experience $X^+$ is encapsulated as the ensemble of isolated layer-wise causal substrates: $\mu^+ = \{W^{(l)+}\}_{l=1}^L$. This structural assembly reveals that engram isolation is equivalent to a **global linear operation** on the parameter manifold, governed by a block-diagonal spectral projector $P^+ = \text{diag}(P^{(1)}, \ldots, P^{(L)})$:

$$\mu^+ = \mu P^+. \quad (5)$$

This formalization establishes a fundamental theoretical result: despite the inherent non-linearity of the network's forward pass, the **functional topology of memory** resides in a linearizable subspace of the weight space. This linearity provides the algebraic foundation for engram compositionality, allowing the zero-shot synthesis or erasure of complex

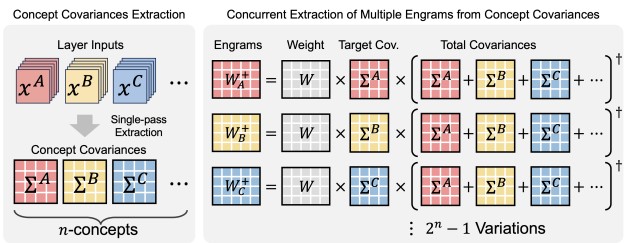

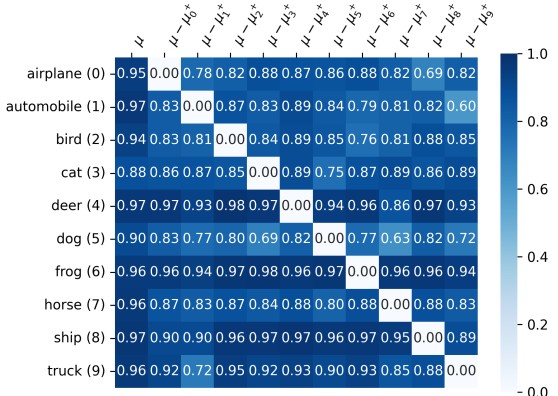

*Figure 3.* **Illustration of Engrammatic Decomposition.** Linear additivity of spectral subspaces enables the zero-shot synthesis of $2^n - 1$ unique memory states from $n$ extracted engrams, mastering the combinatorial explosion of unlearning scenarios (see Appendix F).

knowledge through simple spectral arithmetic. We provide the formal operator-theoretic derivation and proof of this global resolution in Appendix D.

### 4.6. Engrammatic Decomposition: Combinatorial Unlearning

For a pre-trained weight $W$ encoding concepts $\mathcal{C} = \{c_1, \ldots, c_n\}$, we define the individual engram $W_i^+$ by partitioning the total covariance space (see Fig. 3):

$$W_i^+ = W \Sigma_i \left( \sum_{j=1}^{n} \Sigma_j \right)^\dagger. \qquad (6)$$

Since each $\Sigma_i$ is computed independently and remains fixed, we can leverage the **linear additivity** of these components for zero-shot combinatorial unlearning.

For any subset $\mathcal{U} \subseteq \mathcal{C}$, the unlearned state is synthesized via simple linear arithmetic, bypassing the $\mathcal{O}(2^n \cdot \mathcal{T}_{\text{unlearn}})$ complexity of iterative methods with a linear $\mathcal{O}(n \cdot \mathcal{T}_{\text{stat}})$ pre-computation. With this we define **combinatorial unlearning** as:

$$\text{Engram}(\alpha) := W - \alpha \sum_{c_k \in \mathcal{U}} W_k^+, \qquad (7)$$

where $\alpha$ controls the edit strength. This algebraic compositionality enables instant, on-demand unlearning by treating memories as independent, manipulable units.

## 5. Experimental Validation

We validate the proposed engrammatic decomposition through three experiments: **(1) versatility and scalability** across different architectures and datasets, **(2) quantitative comparisons** with state-of-the-art baselines, and **(3) qualitative demonstrations** of engram arithmetic.

### 5.1. Scalability and Versatility: Engram Unlearning

The engrammatic decomposition derived in Eq. (7) enables the simultaneous isolation and causal validation of mul-

*Figure 4.* **Surgical Precision on CIFAR-10 (ResNet-18).** The diagonal drop in accuracy confirms that ablating a specific engram selectively erases the target concept, while the off-diagonal stability indicates that reference classes remain intact.

tiple memory traces without the cumulative overhead of iterative optimization. Our framework **scales** seamlessly to high-dimensional concept spaces, effectively neutralizing the computational burden of massive label sets. It effortlessly accommodates the combinatorial complexity of removing any subset of concepts—whether scaling to the 100 classes of CIFAR-100 or the 1,000 categories of ImageNet-1k (Fig. 12 & 13)—by synthesizing all potential unlearned states from a single, unified linear decomposition.

Furthermore, we demonstrate the **universal versatility** of our framework across a comprehensive architectural spectrum (See Appendix C)—ranging from fundamental MLPs and CNN (ResNet) to modern Vision Transformers (ViT) and Convolutional Autoencoders (ConvAE). In the supervised domain, Fig. 4 confirms that ablating an engram in a ResNet-18 on CIFAR-10 cleanly removes the target class while preserving unrelated concepts. This behavior extends to the unsupervised generative regime: as shown in Fig. 5, removing an engram from a ConvAE selectively impairs the reconstruction of specific morphological features (e.g., a specific digit) even in the absence of explicit labels. These results suggest that the Engram functions as a fundamental, architecture-agnostic unit of memory, effective across both discriminative and generative paradigms.

### 5.2. Quantitative Comparisons: Engram Unlearning

We benchmark Engram against widely used machine unlearning methods on the class-wise unlearning problem. We evaluate on CIFAR-10 and CIFAR-100 using ResNet-18/50, targeting Class 0 for unlearning (See experimental details in Appendix B.4.1). Table 2 summarizes the quantitative comparisons. Engram achieves promising performance in the Tug-of-War (ToW) metric (Zhao et al., 2024), demonstrating superior output-level unlearning. Importantly, this efficacy extends to the latent space; the representation-level metrics

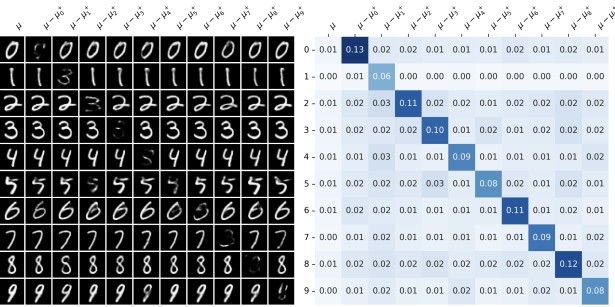

*Figure 5.* **Selective Erasure in ConvAE (MNIST).** Ablating target engrams ($\mu - \mu^+$) selectively impairs reconstruction (Left), as confirmed by the specific increase in test-set MSE (Right).

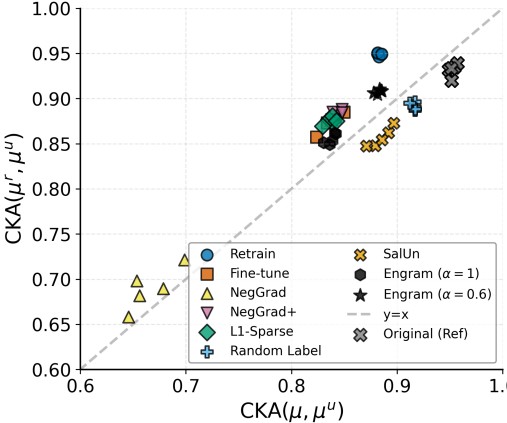

*Figure 6.* CKA similarity of the original (x-axis) versus the re-trained (y-axis) models; ideal unlearning appears toward top-left.

(DA and NMI) (Seo et al., 2025) confirm that our method effectively dissolves the structural traces of the forget set, rather than merely masking the output. (For detailed formulations of these metrics, please refer to Appendix B.4.1.)

We also compute Centered Kernel Alignment (CKA) (Kornblith et al., 2019; Davari et al., 2023; Kim et al., 2026) between unlearned models and both the original and re-trained models. Figure 6 visualizes these relationships: the ideal unlearned model occupies the top-left region, indicating high similarity to the retrained model and low similarity to the original. Engram method with best $\alpha$ consistently positions in this region and is closest to the retrained model.

### 5.3. Qualitative Demonstrations: Engram Arithmetics

**Continuous Semantic Control (Scalar Multiplication).** If the engram $W_c^+$ represents a true causal direction in the weight space, traversing along this vector should yield smooth semantic interpolation. As shown in Fig. 7 (Top), by scaling the engram with a coefficient $\alpha$ (i.e., $\mu' = \mu - \alpha\mu_c^+$), we achieve fine-grained control over attribute intensity (e.g., 'Eyeglasses' or 'Bangs'). This linearity mirrors the property of *Linear Mode Connectivity* (Frankle et al., 2020), suggesting that the engram defines a stable, linearizable trajectory for semantic manipulation without retraining (See more examples in Fig. 15).

**Compositional Vector Arithmetic (Vector Addition).** We further validate the combinatorial nature of these traces through **Engram Arithmetics**. Analogous to the semantic algebra found in word embeddings (e.g., King − Man + Woman ≈ Queen) (Mikolov et al., 2013), we observe that engrams in the weight space exhibit similar additive compositionality. Figure 8 demonstrates that multiple attributes can be simultaneously manipulated via vector addition:

$$\mu_{\text{new}} = \mu \pm \mu_{\text{Glasses}}^+ \mp \mu_{\text{Goatee}}^+. \tag{8}$$

This observation aligns with recent findings in *Task Arithmetic* (Ilharco et al., 2023), confirming that distinct semantic concepts are encoded as orthogonal task vectors that can be linearly superimposed without interference.

### 6. Geometric Analysis of Engram

Our closed-form solution was derived from biological principles via Lagrangian optimization. Remarkably, the same solution emerges from a purely geometric perspective through the Fisher information metric (Amari, 1998). This convergence reveals that Engram is not merely a biologically-inspired heuristic, but captures **a fundamental geometric**

*Table 2.* Class-wise unlearning comparison on CIFAR-10/100. We report ToW(↑), DA(↑), and NMI(↓). Values in parentheses indicate the gap with the retrain models. Best values are **bolded**, and second-best values are underlined. Shaded rows indicate our proposed method and $\alpha_{\text{best}}$ is the best $\alpha$ from grid search.

| CIFAR-10 (RESNET18) | | | |
|---|---|---|---|
| METHOD | ToW↑ | DA↑ | NMI↓ |
| RETRAIN | 0.999 | 0.987 | 0.410 |
| FINE-TUNE | 0.952 (0.047) | 0.973 (0.014) | 0.547 (0.137) |
| NEGGRAD | 0.711 (0.288) | 0.970 (0.017) | 0.037 (0.373) |
| NEGGRAD+ | 0.936 (0.063) | 0.942 (0.045) | 0.244 (0.166) |
| $l_1$-SPARSE | 0.956 (0.043) | 0.975 (0.012) | 0.515 (0.105) |
| RANDOM LABEL | 0.916 (0.083) | 0.851 (0.136) | 0.996 (0.586) |
| SALUN | 0.878 (0.121) | 0.833 (0.154) | 0.911 (0.501) |
| ENGRAM ($\alpha = 1$) | 0.930 (0.069) | **0.992 (0.005)** | **0.379 (0.031)** |
| ENGRAM ($\alpha_{\text{BEST}}$) | **0.984 (0.015)** | 0.958 (0.029) | 0.611 (0.201) |

| CIFAR-100 (RESNET50) | | | |
|---|---|---|---|
| METHOD | ToW↑ | DA↑ | NMI↓ |
| RETRAIN | 0.985 | 0.734 | 0.525 |
| FINE-TUNE | 0.860 (0.125) | 0.768 (0.034) | 0.585 (0.060) |
| NEGGRAD | 0.593 (0.392) | 0.677 (0.057) | 0.149 (0.376) |
| NEGGRAD+ | 0.894 (0.091) | 0.632 (0.102) | 0.120 (0.405) |
| $l_1$-SPARSE | 0.837 (0.148) | 0.770 (0.036) | **0.546 (0.021)** |
| RANDOM LABEL | 0.964 (0.021) | 0.799 (0.065) | 0.778 (0.253) |
| SALUN | 0.980 (0.005) | **0.750 (0.016)** | 0.634 (0.109) |
| ENGRAM ($\alpha = 1$) | 0.988 (0.003) | 0.757 (0.023) | 0.573 (0.048) |
| ENGRAM ($\alpha_{\text{BEST}}$) | **0.983 (0.002)** | 0.767 (0.033) | 0.496 (0.029) |

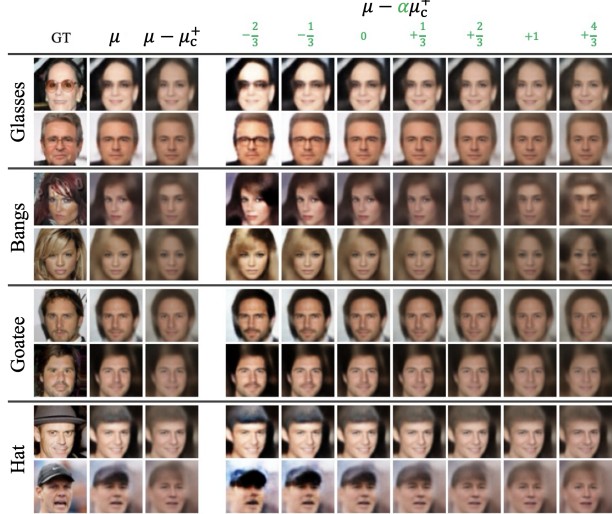

*Figure 7.* **Semantic Specificity Control in WAE (CelebA).** Ablating attribute-specific engrams ($\mu - \mu_c^+$) removes fine-grained target features (e.g., eyeglasses, bangs) while strictly preserving facial identity. The slider ($\mu - \alpha\mu_c^+$) demonstrates the linear compositionality of identified engrams, allowing for fine-grained, continuous manipulation of semantic intensity without retraining.

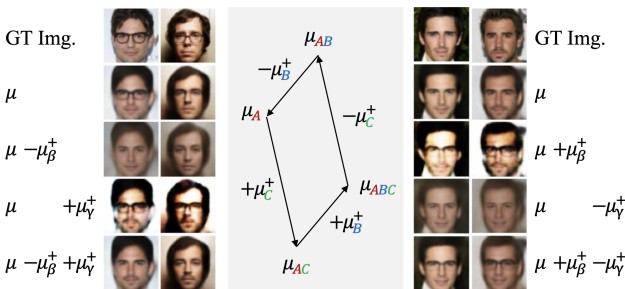

*Figure 8.* **Engram Arithmetics** Examples showing bidirectional arithmetic operation for reconstruction with identified engrams of $\mu_\beta^+$ (Glasses) and $\mu_\gamma^+$ (Goatee). GT, Ground Truth.

**property of neural representations**. In deep neural networks, exact Fisher Information Matrix (FIM) computation is intractable due to high dimensionality. Kronecker Factorization Approximation Curvature (K-FAC) addresses this through two key approximations (Martens & Grosse, 2015).

### 6.1. Fisher and K-FAC Approximation

K-FAC assumes layer independence, treating the FIM as block-diagonal and decomposing curvature layer-wise. This ignores inter-layer correlations but captures essential curvature for optimization (Martens & Grosse, 2015). Within each layer, K-FAC assumes independence between input activations and output gradients. As the FIM block is an expectation of their outer product, this factorizes it into a Kronecker product:

$$F_l \approx A_{l-1} \otimes G_l \qquad (9)$$

where $A_{l-1} = \mathbb{E}[XX^\top]$ captures input activation geometry and $G_l = \mathbb{E}[gg^\top]$ captures output gradient covariance.

Since we analyze a converged model retrospectively, task-specific gradient information $G_l$ is unavailable. Invoking the Principle of Maximum Entropy (Jaynes, 1957), we adopt the least informative prior $G_l \approx \sigma^2 I$—a choice that parallels damping strategies in natural gradient.

### 6.2. Engram as Fisher Projection

**Theorem 6.1** (Fisher-Engram Equivalence). *Let $\mathcal{M}$ be the parameter manifold equipped with the Fisher Information Metric $F$. Under the assumptions of K-FAC ($F_l \approx A_{l-1} \otimes G_l$) and isotropic output curvature (Maximum Entropy Principle, $G_l \approx \sigma^2 I$), Engram identification problem:*

$$\min_{W^+} \mathcal{L}(W^+) = \|(W - W^+)X\|^2 \quad s.t. \quad W^+ X^- = 0 \quad (10)$$

*corresponds to the minimum norm projection onto the soft-constraint manifold under the Fisher metric $F$.*

*Proof.* Under the K-FAC approximation, the global Fisher metric is block-diagonal, treating each layer independently. For a given layer, the local FIM $F_l \in \mathbb{R}^{d_l d_{l-1} \times d_l d_{l-1}}$ decomposes as Eq (9).

Invoking the Principle of Maximum Entropy, we set $G_l \approx \sigma^2 I$. For a given layer with input activations $X$, the input covariance is $A = \mathbb{E}[XX^\top] \approx \frac{1}{N}(\Sigma^+ + \Sigma^-)$, where $\Sigma^+ + \Sigma^- = XX^\top$ is the empirical covariance matrix. Substituting into the K-FAC factorization, the metric tensor simplifies to:

$$F \approx c((\Sigma^+ + \Sigma^-) \otimes I), \quad c = \sigma^2/N$$

On the Riemannian manifold $(\mathcal{M}, F)$, the squared geodesic distance between two points $W$ and $W^+$ is given by the quadratic form induced by the metric tensor:

$$d_F^2(W, W^+) = \text{vec}(W - W^+)^\top F \, \text{vec}(W - W^+)$$

Applying the identity

$$\text{vec}(M)^\top (A \otimes I)\text{vec}(M) = \text{Tr}(MAM^\top)$$

for symmetric matrix A, we observe:

$$d_F^2(W, W^+) = c \, \text{Tr}((W - W^+)(\Sigma^+ + \Sigma^-)(W - W^+)^\top)$$

To establish the connection, consider the Engram objective:

$$\begin{aligned}
\mathcal{L}(W^+) &= \|(W - W^+)X\|^2 \\
&= \text{Tr}((W - W^+)XX^\top(W - W^+)^\top)
\end{aligned}$$

Since $XX^\top = \Sigma^+ + \Sigma^-$, we observe:

$$\text{Tr}((W - W^+)(\Sigma^+ + \Sigma^-)(W - W^+)^\top) \propto d_F^2(W, W^+)$$

Thus, minimizing the Engram loss $\mathcal{L}(W^+)$ is mathematically equivalent to minimizing the Fisher distance $d_F(W, W^+)$ subject to the constraint $W^+ X^- = 0$. □

*Remark* 6.2 (Scope of the Fisher-Engram Equivalence). The isotropic curvature assumption $\boldsymbol{G}_l \approx \sigma^2 \boldsymbol{I}$ is invoked to reveal a *structural equivalence* between two independent derivations—the biological constraints and the Fisher projection—rather than as an operational requirement. The empirical estimator (Eq. (4)) uses input statistics alone and does not invoke this assumption. Relaxing $\boldsymbol{G}_l$ to anisotropic curvature would recover output-side weighting via K-FAC's gradient block, but is unnecessary for the closed-form result.

**Corollary 6.3** (Natural Gradient Interpretation). *Define the forgetting objective $\mathcal{L}_{\text{forget}}(W) = \|W X^+\|_F^2$. Under the K-FAC approximation with isotropic output curvature, the ablated model $W - W^+$ corresponds to a unit-step natural gradient descent on $\mathcal{L}_{\text{forget}}$.*

*Proof.* The Euclidean gradient is $\nabla_W \mathcal{L}_{\text{forget}} = 2W\Sigma^+$. Under K-FAC, the natural gradient becomes:

$$\tilde{\nabla} \mathcal{L}_{\text{forget}} = F^\dagger \nabla \mathcal{L}_{\text{forget}} \propto W\Sigma^+(\Sigma^+ + \Sigma^-)^\dagger \approx W^+.$$

Thus, $W - W^+ \approx W - \tilde{\nabla} \mathcal{L}_{\text{forget}}$, confirming that engram ablation is equivalent to natural gradient forgetting. □

*Remark* 6.4 (Statistical Interpretation). Under the isotropic output curvature assumption ($G_l \approx \sigma^2 I$), the Fisher metric becomes a covariance-weighted norm. Unlike the *Mahalanobis distance* which normalizes via inverse covariance, our metric weights directly by covariance, assigning higher cost to frequent directions to respect the data's statistical structure.

### 6.3. Structure of Effective Projection Operators

We now characterize this structure through spectral analysis of the Effective projection operators.

**Near-orthogonality from Lagrangian structure.** Each concept $\mathcal{C}_i$, defined by target data $X_i^+$ and reference $X_i^-$, induces the operator $P_i = \Sigma_i^+(\Sigma_i^+ + \Sigma_i^-)^\dagger$, where $\Sigma_i^+ = X_i^+ X_i^{+\top}$. Although $P_i$ is not an idempotent projection ($P_i^2 \neq P_i$), it acts as a *soft projection* that effectively isolates the signal subspace based on spectral signal-to-noise ratios. The Lagrangian formulation naturally enforces minimal spectral interference: the minimum-norm solution prefers directions that minimize interference with other memories. This soft-projection structure is consistent with contemporary neuroscience evidence that biological engrams overlap rather than strictly partition (Josselyn & Tonegawa, 2020; Buzsáki, 2004; Kolibius et al., 2025): shared covariance directions between concepts receive attenuated weight rather than binary exclusion, accommodating overlapping but distinguishable memory traces.

*Table 3.* Evaluation results on Llama3.2-1B with TOFU dataset. The top two rows are baselines. Best values among the methods are **bolded**, and second-best values are underlined. $\alpha = 0.6$ denotes uniform $\alpha$ across layers, and $\alpha_{\text{W-Norm}}$ denotes adaptive $\alpha$ obtained by rescaling the weight norm ratio $\|W^+\|/\|W\|$ to $[0, 1]$. EM: Exact Memorization, FQ: $\log_{10}$ Forget Quality.

| METHOD | MEM. ↑ | UTIL. ↑ | PRIV. ↑ | EM ↓ | FQ ↑ |
|---|---|---|---|---|---|
| INIT FINE. | 0.0000 | 1.0000 | 0.0038 | 1.0000 | -21.4083 |
| RETAIN | 1.0000 | 0.9933 | 1.0000 | 0.0000 | 0.0000 |
| GRADDIFF | 0.1849 | **0.9934** | 0.1471 | 0.7182 | -15.8262 |
| IDKNLL | 0.4547 | 0.9102 | 0.0578 | 0.6211 | -13.4845 |
| UNDIAL | 0.5644 | 0.9155 | 0.2272 | 0.7399 | -11.3335 |
| RMU | 0.8660 | 0.7471 | 0.6799 | 0.2953 | -0.2357 |
| ALTPO | 0.9254 | 0.9864 | **0.9549** | 0.0953 | -0.7424 |
| NPO | 0.9339 | 0.9501 | 0.9484 | 0.0948 | -1.9986 |
| SIMNPO | 0.9435 | 0.9706 | 0.9232 | 0.1035 | -0.0897 |
| IDKDPO | 0.9532 | 0.9660 | 0.7751 | 0.1155 | -1.1856 |
| ENGRAM ($\alpha = 0.6$) | 0.9176 | 0.8801 | 0.4832 | **0.0069** | -0.6125 |
| ENGRAM ($\alpha_{\text{W-NORM}}$) | **0.9627** | 0.9256 | 0.6453 | 0.0276 | **-0.0637** |

**Complement subspace.** The projections do not span the entire parameter space: $\sum_i P_i \neq I$. The complement operator $Q := I - \sum P_i$ projects onto the subspace orthogonal to all memory directions.

Perturbations in the range of $Q$ induce zero distance under the degenerate Fisher metric. Among the infinite solutions satisfying the geometric constraints, the Moore-Penrose pseudoinverse implicitly selects the one with the minimum Frobenius norm. This ensures the model avoids parameter drift in these information-null directions, preserving the original weights wherever geometric curvature is absent.

## 7. In Search of Memory Trace in LLM

While Section 6 provides a rigorous geometric foundation, a scaling challenge arises: *can the linear curvature of the Engram capture the truth within the folded, non-linear manifolds of a billion-parameter model?* The dense entanglement of knowledge in Large Language Models (LLMs) often renders global updates destructive. We therefore move beyond theoretical abstraction to test if our closed-form estimator can perform surgical memory isolation on Llama-3.2-1B with TOFU benchmark (See experimental details in Appendix E), seeking the limits of linear-algebraic unlearning in the face of emergent complexity.

**Emergent Performance and the Limits of Uniformity** As evidenced in Table 3, the Engram method demonstrates highest in suppressing *Exact Memorization* metric compared to traditional gradient-based unlearning when employing a **uniform edit strength** ($\alpha = 0.6$). However, the performance gain of established unlearning metrics (Maini et al., 2024) compared to existing baselines is constrained. This suggests that while the linear theory provides the correct "causal direction," the global application with uniform scale at that direction fails to account for the heterogeneous sensi-

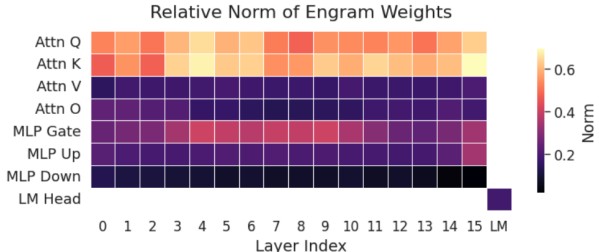

*Figure 9.* **Relative Engram Weight Norm as a Proxy for Causal Memory Traces.** We show the identified relative weight norm of the engram weights over original weight in Llama-3.2-1B for Tofu dataset. The heatmap reveals that the changes are predominantly concentrated in the *Query* and *Key* of self-attention and *Gate* of MLP. With this relative norm, we applied surgical strength in TOFU unlearning process.

tivity across the Transformer layers.

**Engram Weights for Memory Tracing**   To delineate the spatial distribution of memory within the network, we analyzed the relative weight norm ratio W-Norm $= \frac{\|W^+\|_F}{\|W\|_F}$ (Fig. 9). The visualization indicates that memory footprints are not randomly dispersed but display a concentrated structural pattern. The causal substrate is anchored within the **Query** ($W_Q$) and **Key** ($W_K$) projections, alongside the **MLP gating layers** ($gate\_proj$). This empirical evidence identifies the *Memory Trace*, enabling a transition from heuristic manipulation to precise structural intervention. Layer-type ablation in Appendix E.6 (Table 8) confirms this localization causally: Q/K + Gate alone matches the all-layer Overall score, while excluding these layers collapses unlearning performance.

**Structural Automation via W-Norm Scaling**   Based on the identified memory trace, **adaptive scaling ($\alpha_{\text{W-Norm}}$)** adjusts the intervention strength proportional to W-Norm (rescaled to max=1). In Table 3, this adaptive Engram achieves highest memorization while preserving general utility, proving that the model's own spectral statistics provide the optimal schedule for knowledge erasure.

**Engram as a Distinct Operating Point on the Compute–Accuracy Frontier.**   Among closed-form editing methods, Engram outperforms UCE and Task Arithmetic by substantial margins (Table 4); however, it still trails dedicated iterative methods (AltPO, NPO, SimNPO) on Overall and Privacy metrics. This residual gap reflects a structural design choice rather than an empirical shortfall: Engram derives the entire weight update from a single forward pass over target and reference statistics, without backpropagation or iterative refinement, placing it on a fundamentally different point of the compute–accuracy frontier. As detailed in Appendix E.4 (Table 7), gradient-based methods consume roughly $\sim 30\times$ more FLOPs and $\sim 2.3\times$ more peak mem-

*Table 4.* **Comparison with closed-form editing baselines on TOFU** (Llama-3.2-1B, forget10). Engram, UCE, and Task Arithmetic are all gradient-free closed-form methods. Despite this shared paradigm, Engram outperforms both baselines across all metrics except utility. See Appendix E.5 for algebraic correspondence, hyperparameter sweeps, and screening results.

| Method | Overall ↑ | Mem. ↑ | Util. ↑ | Priv. ↑ | EM ↓ |
|---|---|---|---|---|---|
| Task Arithmetic | 0.584 | 0.900 | 0.678 | 0.392 | 0.128 |
| UCE | 0.659 | 0.901 | 0.951 | 0.418 | 0.135 |
| **Engram** | **0.818** | **0.963** | 0.926 | **0.645** | **0.028** |

ory per concept, with wall-clock times on the order of tens of minutes versus $\sim 2$ minutes for Engram. Despite this cost asymmetry, Engram still dominates on Exact Memorization (EM: **0.028** vs. $\geq 0.095$) and remains competitive on Forget Quality.

We therefore position Engram not as a replacement for iterative unlearning but as an orthogonal regime: $\mathcal{O}(1)$-pass, gradient-free memory isolation that complements rather than competes with refinement-based methods, particularly valuable when (i) the number of concepts to unlearn is large (combinatorial compositionality, Section 4.6), (ii) compute or weight access is restricted, or (iii) the goal is structural identification rather than distributional indistinguishability. We discuss further scope conditions and the interpretive distinction between functional identification and learning-trajectory reconstruction in Appendix H.

## 8. Conclusion

We introduced a principled framework for identifying and manipulating memory traces in artificial intelligence. By formalizing neuroscientific engram axioms into a constrained inverse problem, we have derived a scalable, closed-form estimator that remains robust across diverse architectures—from MLP, CNN, and ViT to billion-parameter LLM.

Our theoretical analysis establishes a theoretical link between biological memory axioms and information geometry, specifically through the lens of **Fisher-KFAC approximation**. Beyond its geometric view, the empirical discovery of the condensed localization pattern of **memory traces** in the LLM provides a new perspective on neural interpretability. We have demonstrated that even within highly non-linear systems, causal substrates of knowledge can be isolated and manipulated through deterministic spectral resolutions.

By shifting the paradigm of machine unlearning from *stochastic gradient search* to *structural identification*, our framework enables surgical memory isolation with unprecedented efficiency. Future work may examine the temporal dynamics of engrams in continual learning and find the bridge between biological memory theories and the pursuit of robust, interpretable artificial intelligence.

## Impact Statement

This work identifies AI engrams, causal memory units that allow learned knowledge to be composed or erased through simple linear arithmetic. By enabling precise editing without retraining, our method could support safer model behavior and clearer interpretability. These results establish a novel connection between theories of biological memory and artificial representation learning, pointing toward more accountable and controllable AI systems. As models become easier to edit at the level of individual memories, a broader discussion is needed about how and by whom such fine-grained control should be governed.

## Acknowledgment

This work was partly supported by the National Science Foundation of Korea (NRF) Grant [RS-2022-00165347, RS-2026-25469757]. We thank Krishna P. Gummadi and Peter Dayan for valuable discussions and encouraging feedback on this work.

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

# A. Derivation of Closed-form Solution

## A.1. Constrained Optimization Formulation

From the engram objective function Eq. 2 established by neuroscience engram criteria, we now formulate the optimization problem.

**Proposition A.1** (**Engram Identification Objective**). *The optimal synaptic engram $W^+$ is the solution to the following constrained optimization problem:*

$$\underset{W^+}{minimize} \quad \mathcal{J}(W^+) = \frac{1}{2}\left\|(\Delta W - W^+)X^+\right\|_F^2 \quad (11)$$

$$subject\ to \quad W^+X^- = 0. \quad (12)$$

This mathematical formulation directly maps to two biological criteria: Eq.(11) satisfies **causal reactivation** by maximizing the target's reconstruction fidelity, whereas Eq.(12) enforces **target specificity** by guaranteeing zero interference with reference memories.

*Remark* A.2 (**Isolation of Unique Traces**). *Strict orthogonality* is an analytical instrument to isolate the **unique causal component** of the target memory. Although neural networks naturally rely on shared entangled representations, this formulation mathematically removes the shared substrate, leaving only the specific memory trace that separates the target $X^+$ from the background knowledge $X^-$.

This formulation yields a trace is causal for the target (minimizing the reconstruction error) and inert with respect to the reference (null-space constraint).

## A.2. Lagrangian Derivation and KKT Conditions

To obtain a closed-form solution to the constrained optimization problem, we introduce a Lagrange multiplier matrix $\Lambda \in \mathbb{R}^{m \times n}$ corresponding to the equality constraint (12). The Lagrangian function $\mathcal{L}(W^+, \Lambda)$ is given by:

$$\mathcal{L} = \frac{1}{2}\operatorname{tr}\left(X^{+\top}(\Delta W - W^+)^\top(\Delta W - W^+)X^+\right)$$
$$- \operatorname{tr}\left(\Lambda^\top W^+ X^-\right). \quad (13)$$

The optimal solution is driven by the Karush–Kuhn–Tucker (KKT) conditions, which provide stationary points of equality-constrained problems.

**1. Stationarity:** We enforce this condition by setting the gradient of the Lagrangian with respect to $W^+$ to zero:

$$\nabla_{W^+}\mathcal{L} = -(\Delta W - W^+)X^+X^{+\top} - \Lambda X^{-\top} = 0. \quad (14)$$

**2. Primal Feasibility:** The solution must also satisfy the original constraint:

$$W^+X^- = 0. \quad (15)$$

To solve for $W^+$, we turn KKT into a projection problem: the engram must reconstruct the observed weight update on the target subspace while projecting the reference subspace to zero. Introducing the concatenated matrix $\mathcal{X} := [X^+ \ X^-]$ allows us to combine reconstruction and orthogonality into a unified block linear projection:

$$W^+\mathcal{X} = \begin{bmatrix} \Delta W X^+ & 0 \end{bmatrix}. \quad (16)$$

The general solution to Eq. (16) is given by the Moore-Penrose pseudoinverse $\mathcal{X}^\dagger$, yielding a closed-form expression with clear geometric meaning:

$$W^+ = \begin{bmatrix} \Delta W X^+ & 0 \end{bmatrix}\mathcal{X}^\dagger. \quad (17)$$

## A.3. Scalable Estimation via Sufficient Statistics

Directly computing the pseudoinverse of the **concatenated input matrix** $\mathcal{X} \in \mathbb{R}^{d \times N}$ in Eq. (17) is computationally prohibitive for large datasets, since its size grows linearly with sample size $N$. To make this tractable, we reformulate the solution by exploiting the structure of the pseudoinverse.

Using the identity $\mathcal{X}^\dagger = \mathcal{X}^\top(\mathcal{X}\mathcal{X}^\top)^\dagger$, we can express the solution in terms of the **total covariance matrix** $\mathcal{X}\mathcal{X}^\top$. Substituting this into Eq. (17) yields:

$$W^+ = \begin{bmatrix} \Delta W X^+ & 0 \end{bmatrix}\underbrace{\begin{bmatrix} X^{+\top} \\ X^{-\top} \end{bmatrix}}_{\mathcal{X}^\top}\underbrace{(\mathcal{X}\mathcal{X}^\top)^\dagger}_{\text{Inv. Covariance}}$$
$$= \left(\Delta W X^+ X^{+\top} + 0\right)\left(X^+X^{+\top} + X^-X^{-\top}\right)^\dagger. \quad (18)$$

This derivation reveals that the entire computation depends on the uncentered covariance matrices, defined as:

$$\Sigma^+ = X^+(X^+)^\top, \quad \Sigma^- = X^-(X^-)^\top. \quad (19)$$

These covariance terms are sufficient statistics: once computed, the full dataset is no longer required.

**Computational Advantage.** Because the covariance matrices are additive, they can be **accumulated iteratively** over mini-batches. This avoids storing the full activation matrix and updates statistics on the fly: $\Sigma \leftarrow \Sigma + X_{\text{batch}}X_{\text{batch}}^\top$. The memory footprint drops from linear $\mathcal{O}(Nd)$ to a constant $\mathcal{O}(d^2)$. Such scalability is essential for analyzing modern large-scale models where $N \gg d$.

Substituting these statistics into the expanded equation gives our final, scalable closed-form solution:

$$\boxed{W^+ = \Delta W \, \Sigma^+ \left(\Sigma^+ + \Sigma^-\right)^\dagger.} \quad (20)$$

By utilizing the pseudoinverse (†), this solution inherently provides the minimum-norm solution when the covariance matrix is rank-deficient (e.g., due to dead neurons), avoiding the need for manual hyperparameter tuning.

### A.4. Practical Instantiation: Retrospective Estimation

The theoretical derivation uses the differential update $\Delta \boldsymbol{W}$, but applying this framework to deep networks requires reconsidering the role of initialization. As defined in Section 3, the initial state $\boldsymbol{\mu}_0$ serves as a *functional baseline*. In deep models, however, the random initialization $\boldsymbol{W}_0$ acts as a high-entropy state with no coherent response to the shifted input manifold $\boldsymbol{X}_{t=1}$. Therefore, preserving it as a reference would introduce noise rather than a meaningful baseline.

Consistent with defining learning as a gain-of-function from a null state, we instantiate the effective memory update as the total accumulated functional structure:

$$\Delta \boldsymbol{W}_{\text{effective}} \approx \boldsymbol{W}_{\text{trained}} - \mathbf{0}. \tag{21}$$

Under this framework, the condition of *necessity* implies **functional silencing**: removing the engram should return the system to a null-response state for the target, not to the random fluctuations of initialization. Substituting this practical form into Eq. (20) yields the final estimator:

$$\boldsymbol{W}^{+} = \boldsymbol{W}_{\text{trained}} \, \boldsymbol{\Sigma}^{+} \left( \boldsymbol{\Sigma}^{+} + \boldsymbol{\Sigma}^{-} \right)^{\dagger}. \tag{22}$$

This formulation enables the rigorous engram extraction using only the final model checkpoint and current activation statistics, fully decoupling identification from the training trajectory while satisfying the causal definitions.

## B. Experimental Details for Causal Validation of Engram Method

In this section, we provide the comprehensive technical specifications, architectural configurations, and hyperparameter settings utilized to validate the **AI Engram** framework. Our experiments are designed to demonstrate the framework's versatility across discriminative, generative, and large-scale foundational models.

### B.1. Computational Resources

All engram extraction processes were performed on a single NVIDIA A100 (80GB) GPU. Due to the closed-form nature of the estimator, the extraction of all 100 class engrams for CIFAR-100 (ResNet-18) takes less than 2 minutes, compared to several hours required for iterative gradient-based unlearning methods.

### B.2. Architectural Configurations and Datasets

We evaluate the **Spectral AI Engram** framework on four primary benchmarks, encompassing both discriminative and generative tasks across various parameter scales:

- **CIFAR-10/100 (ResNet-18):** We utilize a standard ResNet-18 architecture (He et al., 2016) pre-trained on CIFAR-10 and CIFAR-100. The models are sourced from the Hugging Face Hub (e.g., `edadaltocg/resnet18_cifar10/100`), achieving 95.2% top-1 accuracy on CIFAR-10 and 77.8% on CIFAR-100 respectively. For these discriminative tasks, engrams are extracted from the convolutional filters and final linear projection layers.

- **MNIST (ConvAE):** A 3-layer Convolutional Autoencoder is implemented. The encoder employs $3 \times 3$ kernels with 16 and 32 channels, leading to a latent bottleneck of 64 dimensions. The model is optimized using MSE loss and the Adam optimizer for 50 epochs to establish a baseline for morphological reconstruction fidelity.

- **CelebA (WAE):** A Wasserstein Autoencoder (WAE) (Tolstikhin et al., 2018) is utilized to investigate latent manifold editing. The architecture follows a DCGAN-based structure with a latent dimension of $d = 128$ and an MMD penalty. The model is trained on cropped CelebA images ($64 \times 64$) to encode complex semantic attributes, enabling the validation of engram arithmetic.

- **ImageNet-1K (ViT-B/16):** To verify the scalability of our $O(d^2)$ estimator, we employ a pre-trained Vision Transformer (`google/vit-base-patch16-224`) (Dosovitskiy et al., 2021) from the Hugging Face Transformers library. Input images are processed via the

  `ViTImageProcessor` with RGB conversion and a resolution of $224 \times 224$. We focus on the internal projection layers ($W_Q, W_K, W_V, W_O$) within the transformer blocks and the MLP heads to isolate causal engrams at scale.

### B.3. Engram Extraction: Implementation Details

The extraction process follows the **Retrospective Estimation** protocol described in Section 4.5. Unlike gradient-based unlearning, our method is deterministic and one-shot.

**Covariance Accumulation.** The sufficient statistics $\boldsymbol{\Sigma}^{+}$ and $\boldsymbol{\Sigma}^{-}$ are computed during a single forward pass over the target ($X^{+}$) and reference ($X^{-}$) datasets. To maintain a constant memory footprint, we accumulate the uncentered covariance matrices iteratively:

$$\boldsymbol{\Sigma} \leftarrow \boldsymbol{\Sigma} + \sum_{i=1}^{B} \mathbf{x}_i \mathbf{x}_i^{\top} \tag{23}$$

where $B$ is the mini-batch size. For all experiments, we use $B = 128$. This reduces the space complexity from $O(Nd)$ to $O(d^2)$, where $d$ is the layer dimension (e.g., $d = 768$ for ViT-B).

**Numerical Stability and Pseudoinverse.** To compute $(\Sigma^+ + \Sigma^-)^\dagger$, we employ the Moore-Penrose pseudoinverse using Singular Value Decomposition (SVD). To ensure numerical stability against "dead neurons" or rank-deficient subspaces, we apply a singular value threshold $\epsilon = 10^{-6} \times \lambda_{max}$. This provides the **minimum-norm solution** for $W^+$, which is critical for maintaining model stability after ablation.

### B.4. Evaluation Protocols for Unlearning

#### B.4.1. CIFAR10/100 CLASS-WISE UNLEARNING

We evaluate on CIFAR-10 (10 classes) and CIFAR-100 (100 classes) using ResNet-18 and ResNet-50 architectures. For all experiments, we target Class 0 for unlearning: the model is first trained on the complete dataset, then unlearned to forget Class 0 while retaining knowledge of remaining classes. **Retain set** $X^-$ is training samples from classes other than target class. **Forget set** $X^+$ is training samples from target class. **Test set** $\mathcal{D}_{\text{test}}$ is full test set.

Let $\mu$ denote model parameters, $X^-$ the retain set, $X^+$ the forget set, and $\mathcal{L}_{\text{CE}}$ the cross-entropy loss. Following is a list of the machine unlearning algorithms demonstrated in this work.

**Fine-tune (Warnecke et al., 2023).** Standard fine-tuning on the retain set:

$$\mathcal{L}_{\text{Finetune}} = \mathbb{E}_{(x,y)\sim X^-}\left[\mathcal{L}_{\text{CE}}(f_\mu(x), y)\right]$$

**NegGrad (Thudi et al., 2022).** Gradient ascent on the forget set to maximize loss:

$$\mathcal{L}_{\text{NegGrad}} = -\mathbb{E}_{(x,y)\sim X^+}\left[\mathcal{L}_{\text{CE}}(f_\mu(x), y)\right]$$

**NegGrad+ (Kurmanji et al., 2023).** Combined objective balancing retention and forgetting:

$$\mathcal{L}_{\text{NegGrad+}} = \beta\,\mathbb{E}_{(x,y)\sim X^-}\left[\mathcal{L}_{\text{CE}}(f_\mu(x), y)\right]$$
$$- (1-\beta)\,\mathbb{E}_{(x,y)\sim X^+}\left[\mathcal{L}_{\text{CE}}(f_\mu(x), y)\right]$$

**$l_1$-Sparse (Jia et al., 2023).** Fine-tuning with $l_1$ regularization to promote parameter sparsity:

$$\mathcal{L}_{l_1} = \mathbb{E}_{(x,y)\sim X^-}\left[\mathcal{L}_{\text{CE}}(f_\mu(x), y)\right] + \gamma\sum_i |w_i|$$

**Random Label (Golatkar et al., 2020).** Training on combined data with randomized labels for the forget set:

$$\mathcal{L}_{\text{Rand}} = \mathbb{E}_{(x,y)\sim X^-}\left[\mathcal{L}_{\text{CE}}(f_\mu(x), y)\right]$$
$$+ \mathbb{E}_{(x,\cdot)\sim X^+}\left[\mathcal{L}_{\text{CE}}(f_\mu(x), y_{\text{rand}})\right]$$

where $y_{\text{rand}}$ denotes uniformly sampled incorrect labels.

**SalUn (Fan et al., 2024).** Saliency-based unlearning with masked parameter updates:

1. Compute gradient magnitude $|\nabla_\mu\mathcal{L}_{\text{CE}}(X^+)|$ on the forget set

2. Generate binary mask $M \in \{0,1\}^{|\mu|}$ preserving top $(1-\text{sparsity\_ratio})\%$ parameters by gradient magnitude

3. Update parameters using Random Label objective with mask: $\mu_{t+1} = \mu_t - \eta(M \odot \nabla\mathcal{L}_{\text{Rand}})$

**Engram (Ours).** Closed-form projection that removes the statistical trace of forget data from model weights. For a weight matrix $W$, given input covariance matrices $\Sigma_f$ (forget set) and $\Sigma_{\text{total}} = \Sigma_r + \Sigma_f$ (total training set):

$$W_{\text{new}} = W - \alpha\,W\Sigma_f\Sigma_{\text{total}}^\dagger$$

where $(\cdot)^\dagger$ denotes the Moore-Penrose pseudo-inverse and $\alpha$ controls edit strength. This operation is applied layer-wise to all layers.

**Hyperparameter Search** We perform grid search for each algorithm, selecting hyperparameters that maximize the Tug-of-War (ToW) metric. All methods use batch size 128 with SGD optimizer (momentum 0.9, weight decay $5 \times 10^{-4}$). See Table 5. The best hyperparameter for Engram, $\alpha_{\text{best}}$, obtained from grid search is 0.6 and 1.6 for CIFAR-10 and CIFAR-100, respectively.

*Table 5.* Hyperparameter search spaces for each unlearning method.

| Method | Hyperparameters (LR / Epochs / Others) |
|---|---|
| Fine-tune | LR: {0.1, 0.01, 0.001, 5e-4, 3e-4, 1e-4, 5e-5}, Epochs: 10 |
| $l_1$-Sparse | LR: {as above}, $\gamma$: {1e-6, 1e-5, 1e-4}, Epochs: 10 |
| NegGrad | LR: {as above}, Epochs: 10 |
| NegGrad+ | LR: {as above}, $\beta$: 0.99, Epochs: {5, 10} |
| SalUn | LR: {as above}, Sparsity: 0.5, Epochs: 10 |
| Random Label | LR: {as above}, Epochs: 10 |
| Engram | $\alpha$ (edit strength): {0.5, 0.6, ..., 2.0} (step 0.1) |

**Evaluation Metrics** We adopt the Tug-of-War (ToW) metric (Zhao et al., 2024), which quantifies alignment between an unlearned model $\mu^u$ and a model retrained from scratch $\mu^r$:

$$\text{ToW} = (1 - \text{da}(\mu^u, \mu^r, X^+)) \cdot (1 - \text{da}(\mu^u, \mu^r, X^-))$$
$$\cdot (1 - \text{da}(\mu^u, \mu^r, \mathcal{D}_{\text{test}})),$$

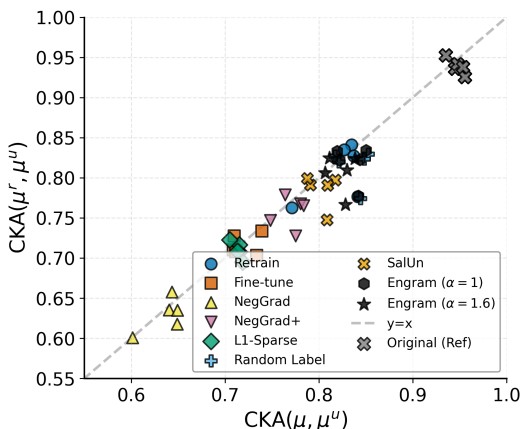

*Figure 10.* CKA similarity of the original (x-axis) versus the retrained (y-axis) models for CIFAR-100.

where $da(\mu^u, \mu^r, \mathcal{D}) = |a(\mu^u, \mathcal{D}) - a(\mu^r, \mathcal{D})|$ denotes the absolute accuracy difference on dataset $\mathcal{D}$. ToW ranges from 0 to 1; higher values indicate closer behavioral alignment to exact unlearning.

Table 2 presents quantitative comparisons. Engram achieves the highest ToW scores on both CIFAR-10 (0.984) and CIFAR-100 (0.983), outperforming all baselines. The closed-form solution of Engram provides an optimal trade-off between forgetting quality, retention, and model utility without iterative gradient optimization.

ToW measures output-level behavior, but models may retain information about the forget set in internal representations without manifesting it at the output layer. We therefore evaluate representation-level metrics (Seo et al., 2025): Dimensional Alignment (DA), which measures how well forget-set features align with the retain-set principal subspace, and Normalized Mutual Information (NMI), which quantifies clustering separability between forget and retain features. Table 2 shows that Engram achieves moderate DA and NMI values, indicating that it retains less information regarding the forget set.

In addition, we compute Centered Kernel Alignment (CKA) (Kornblith et al., 2019; Davari et al., 2023; Kim et al., 2026) between unlearned models and both the original and retrained models. Figure 6 (CIFAR-10) and Figure 10 (CIFAR-100) visualizes these relationships: the ideal unlearned model occupies the top-left region, indicating high similarity to the retrained model and low similarity to the original. Engram method with best $\alpha$ consistently positions in this region and is closest to the retrained model.

**Evaluation Protocol** After hyperparameter selection, we run each unlearning method with 5 different random seeds. Both the original trained model and the retrained model use corresponding seeds for fair comparison.

**Note on Reference Model Metrics.** The retrained model does not achieve ToW = 1.0 because we compute ToW between retrained models with different random seeds (e.g., ToW between retrain-seed-1 and retrain-seed-2). This procedure captures inherent stochasticity in training and provides a realistic upper bound. The same protocol applies to CKA measurements: CKA between retrained models with different seeds is less than 1.0, and CKA for the original model reflects comparison across seeds rather than self-similarity.

## C. Surgical Validation of Engram Method.

### C.1. MNIST on MLP

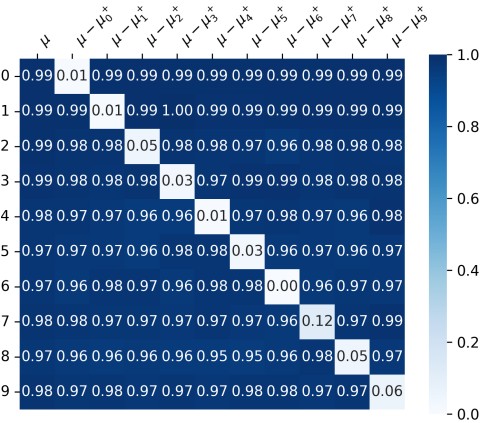

*Figure 11.* **Class-wise unlearning performance on MNIST 3-layer MLP** We evaluate the Engram Method by unlearning each of the 10 classes individually. The heatmap illustrates that the accuracy of the target class (white diagonal) drops to near zero, while the performance on the remaining 99 classes (dark blue off-diagonal) is perfectly preserved, showcasing the method's ability to handle high-dimensional class sets without collateral interference.

## C.2. ImageNet-1k on ViT

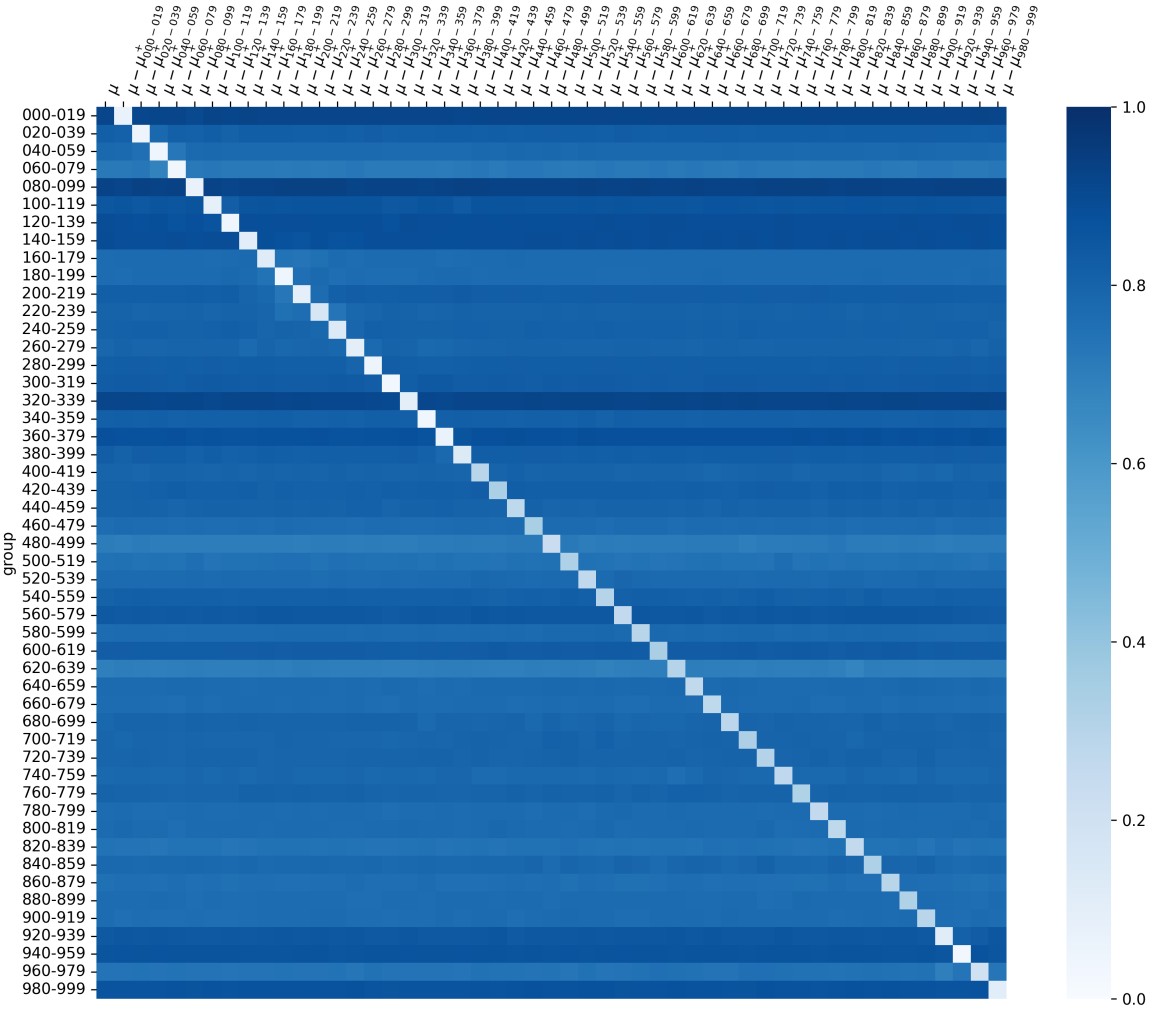

*Figure 12.* **Grouped class-wise unlearning performance on ImageNet-1k with ViT** We show 20-class grouped-wise unlearning result. Each row represents a model trained to unlearn a specific group of classes. The white diagonal elements indicate near-zero accuracy for the target forgotten groups, while the dark blue off-diagonal regions demonstrate that the accuracy for other classes is preserved, confirming the surgical precision of the method.

## C.3. CIFAR-100 on ResNet-18

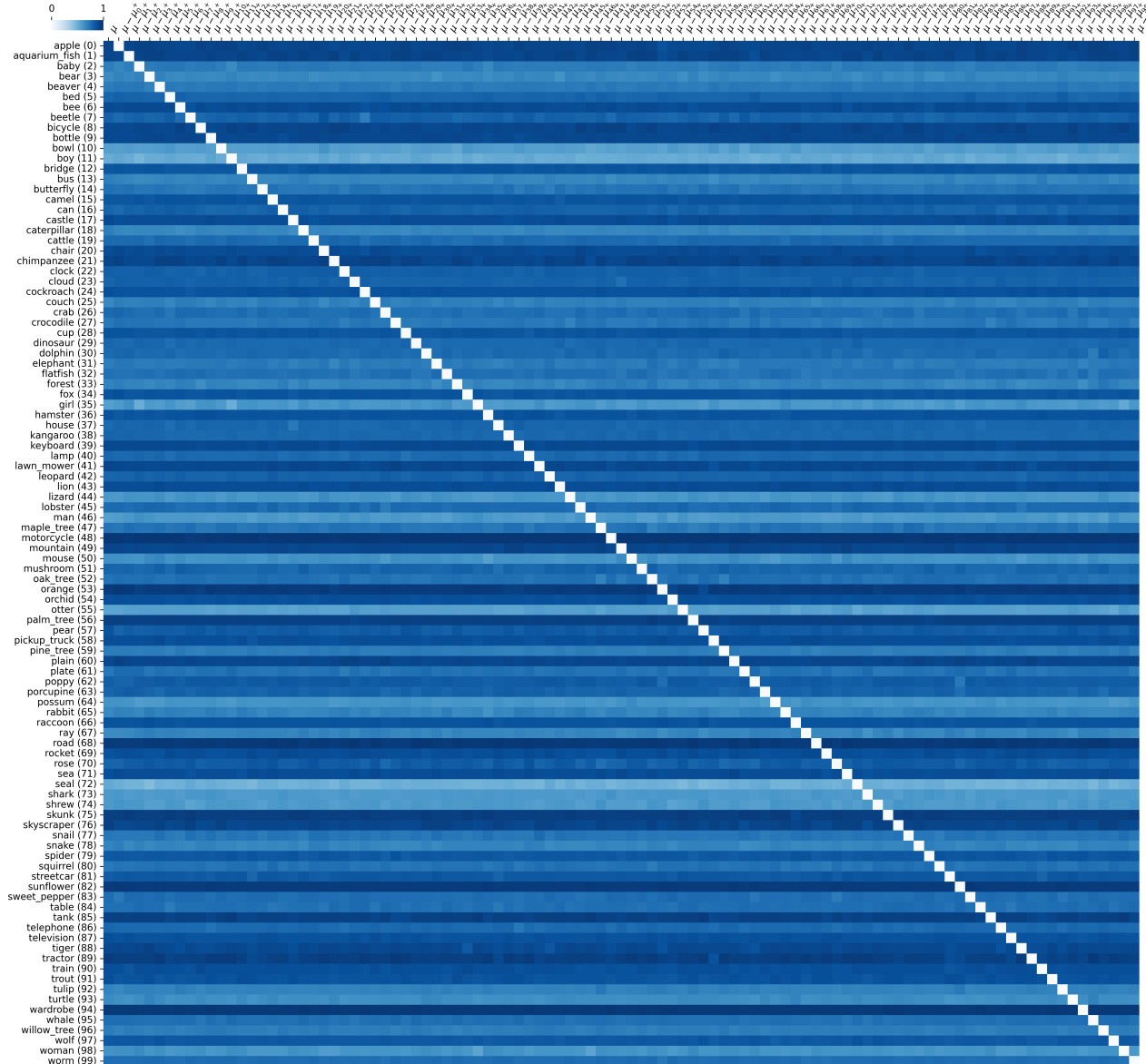

*Figure 13.* **Class-wise unlearning performance on CIFAR-100 in ResNet-18.** Each row $i$ represents a model specifically updated to unlearn class $i$ using the Engram Method. The prominent white diagonal indicates that the accuracy for the target forgotten class drops to near zero, while the consistently dark blue off-diagonal regions demonstrate that the knowledge of all other 99 classes remains intact. This highlights the method's ability to perform class-specific erasure with minimal collateral damage to unrelated categories.

## C.4. Graceful Degradation under Semantic Overlap

The soft-projection structure analyzed in Section 6.3 predicts that engram ablation should degrade smoothly and proportionally with semantic similarity between target and reference concepts, rather than catastrophically failing when the null-space constraint is violated. We empirically verify this prediction on CIFAR-100 with ResNet-18, exploiting the dataset's native two-level structure (20 superclasses, each containing 5 fine classes).

**Setup.** For each fine class $c$ as the target, we extract the engram $W_c^+$ and ablate it from the trained model. We then measure the test-set accuracy drop on every other class $c'$, partitioning $c'$ into two groups: *same-superclass* pairs (high feature overlap, e.g., maple_tree/oak_tree) and *different-superclass* pairs (low overlap, e.g., maple_tree/dolphin). Within each group, we additionally compute cosine similarity between class-conditional mean input representations.

**Results.** Same-superclass pairs show a mean accuracy drop of **0.80** percentage points, while different-superclass pairs show near-zero degradation. Within a superclass, the drop varies smoothly and monotonically with cosine similarity (Fig. 14): semantically closer pairs show proportionally larger interference, but the curve is continuous with no discontinuous collapse. This confirms that the estimator *maximizes* separation rather than *forcing* it—the Lagrangian's hard constraint $W^+ X^- = 0$ shapes the optimization, while the resulting operator $P^+$ accommodates partial concept overlap through its spectral structure.

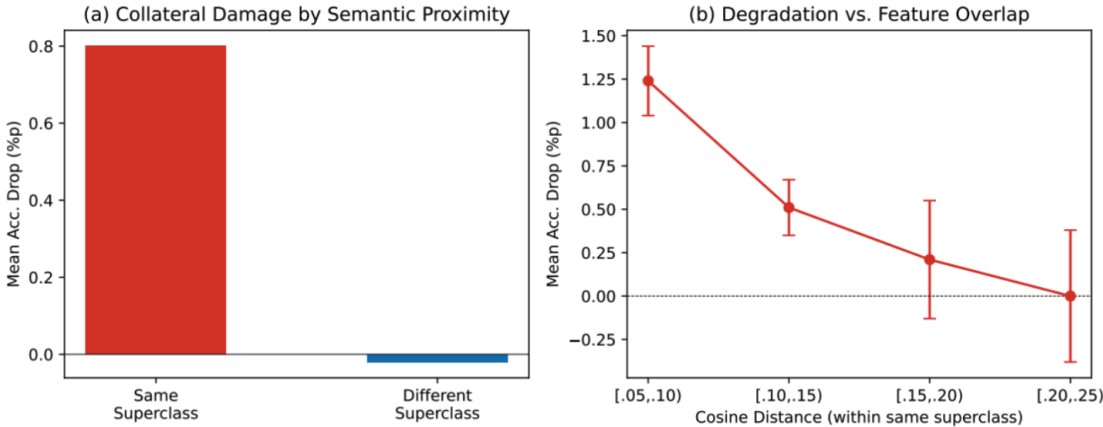

*Figure 14.* **Graceful degradation under semantic overlap on CIFAR-100 (ResNet-18).** (a) Mean test-set accuracy drop on reference classes after ablating the target engram, split by same-superclass (high feature overlap) versus different-superclass (low overlap) pairs. Same-superclass pairs show only 0.80 percentage-point mean degradation; different-superclass pairs are effectively unaffected. (b) Within a superclass, accuracy drop varies smoothly and monotonically with cosine similarity between class-conditional input representations, with no discontinuous collapse. These results empirically confirm that the soft-projection operator $P^+$ accommodates partial concept overlap.

## C.5. Generative Models

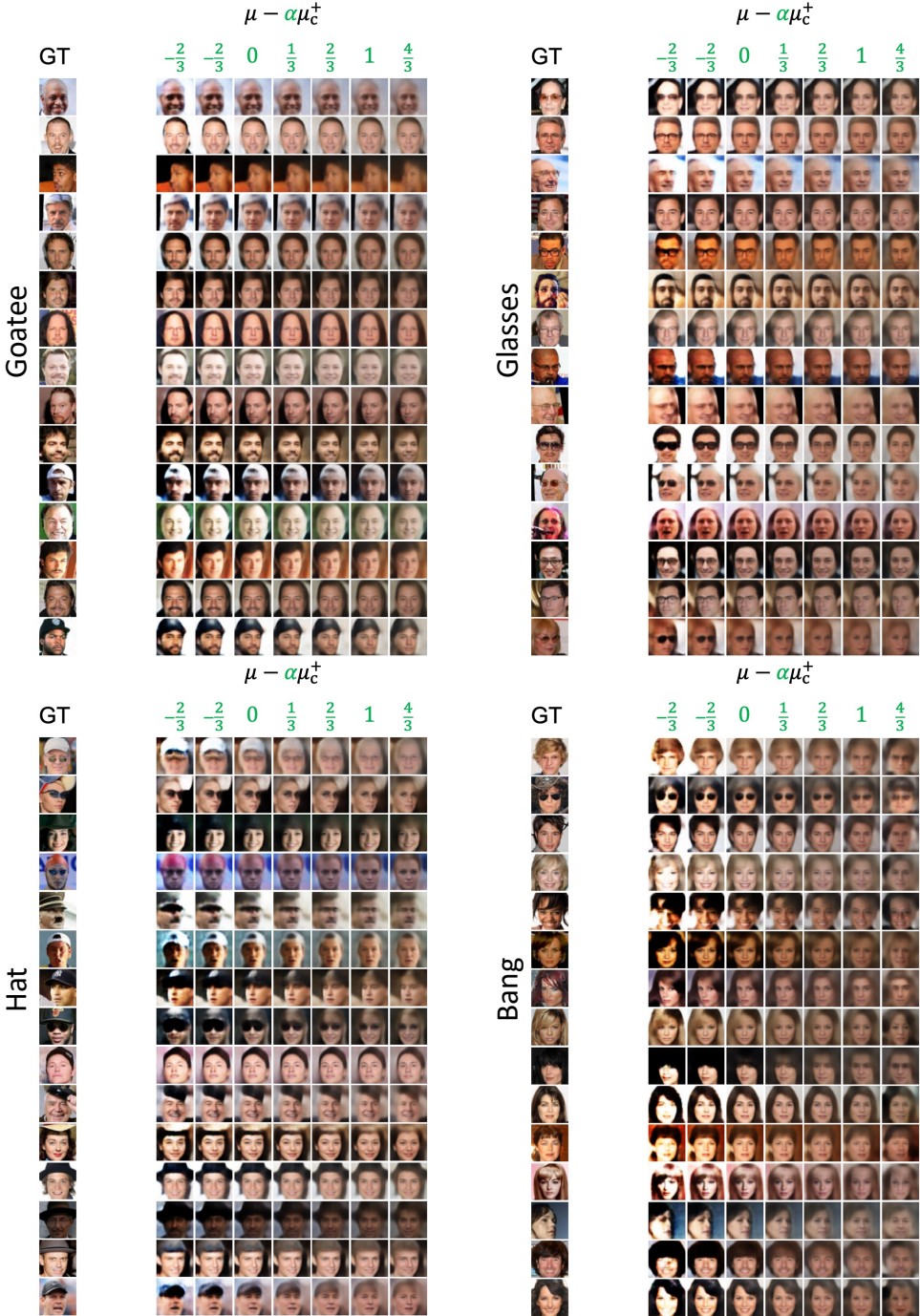

*Figure 15.* **Semantic attribute control on CelebA (WAE).** For each attribute (Goatee, Glasses, Hat, Bangs), we apply continuous engram-based editing $\mu - \alpha\mu_c^+$ with $\alpha \in \{-2/3, -1/3, 0, 1/3, 2/3, 1, 4/3\}$. Increasing $\alpha$ progressively removes the target attribute while preserving facial identity; negative $\alpha$ amplifies the attribute. The smooth, monotonic interpolation across $\alpha$ values demonstrates that the identified engram defines a stable, linearizable trajectory in weight space for semantic manipulation without retraining. GT: ground truth reconstruction at $\alpha = 0$.

## D. Operator-Theoretic Formalism of AI Engrams

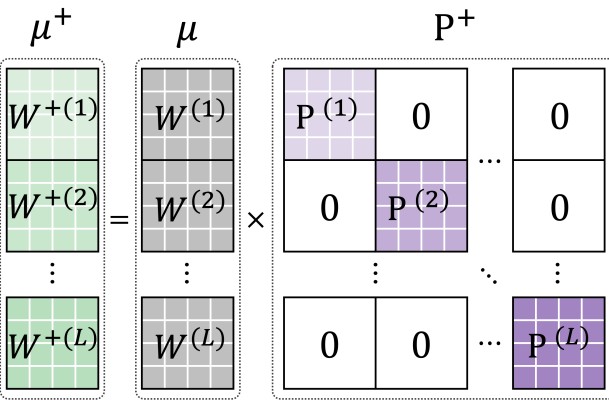

*Figure 16.* Global Engram Operator. Block-diagonal projector P acting on layer-wise weights $\mu$.

To formalize the global isolation of memories, we define a **Global Engram Operator** acting on the parameter manifold $\mathcal{W}$. This formalism elucidates how local synaptic changes aggregate into a global functional structure. Let the global parameter set $\mu$ be represented as a block-column vector of layer-wise weights, and let $\mathbf{P}$ be the corresponding block-diagonal spectral projector:

$$\mu = \begin{bmatrix} \mathbf{W}^{(1)} \\ \mathbf{W}^{(2)} \\ \vdots \\ \mathbf{W}^{(L)} \end{bmatrix}, \quad \mathbf{P}^+ = \text{diag}\left(\mathbf{P}^{+(1)}, \mathbf{P}^{+(2)}, \ldots, \mathbf{P}^{+(L)}\right) \tag{24}$$

where each block $\mathbf{P}^{+(l)} = \Sigma_l^+ (\Sigma_l^+ + \Sigma_l^-)^\dagger$ denotes the spectral filter for layer $l$. As illustrated in Figure 16 the total AI engram $\mu^+$ is the result of a **global linear transformation** in weight space:

$$\mu^+ = \mu \mathbf{P}^+ \tag{25}$$

**Linearity and Global Decoupling.** The block-diagonal structure of $\mathbf{P}$ provides the mathematical justification for our layer-wise decomposition. It proves that identifying the global engram is equivalent to a parallelizable resolution of identity across the weight manifold. Crucially, this confirms that engram isolation is not a heuristic localization but a **spectral resolution of the entire network state**, allowing for the exact reconstruction of causal traces without iterative feedback or inter-layer communication.

## E. Experimental Setup for Large-Language Model Unlearning

### E.1. Experimental Setup

We utilized the **OpenUnlearning** framework (Dorna et al., 2025) to conduct a standardized evaluation of unlearning methods.

- **Dataset:** We employed the **TOFU** (Task-Specific Forgettable Unlearning) benchmark (Maini et al., 2024). Specifically, we focused on the `forget10` split, which targets the unlearning of 10% of the training data ($\mathcal{D}_{forget}$), while retaining the remaining 90% ($\mathcal{D}_{retain}$).

- **Model:** We used the **Llama-3.2-1B-Instruct** model (Grattafiori et al., 2024) as the base model. The initial finetuned model ($\theta_{ft}$) was obtained by full-parameter fine-tuning on the full TOFU dataset ($\mathcal{D}_{forget} \cup \mathcal{D}_{retain}$).

- **Baselines:** We compared our method against a comprehensive set of state-of-the-art baselines:
  - **GradDiff** (Liu et al., 2025): A gradient-based method that maximizes loss on $\mathcal{D}_{forget}$ while minimizing it on $\mathcal{D}_{retain}$.
  - **IdkNLL & IdkDPO** (Maini et al., 2024): Standard baselines that train the model to respond with "I don't know" for forget-set queries using NLL and DPO objectives, respectively.
  - **UNDIAL** (Dong et al., 2025): A self-distillation approach that utilizes adjusted logits for robust knowledge erasure.
  - **RMU** (Li et al., 2024): A representation-based method that misaligns forget-set activations with a steering vector.
  - **NPO** (Zhang et al., 2024): Negative Preference Optimization, which uses a preference-based loss to steer the model away from $\mathcal{D}_{forget}$.
  - **SimNPO** (Fan et al., 2025): A simplified, reference-free variant of NPO designed to mitigate reference model bias.
  - **AltPO** (Mekala et al., 2025): An alternate preference optimization method that incorporates alternate positive labels during unlearning.

### E.2. Evaluation Metrics

We evaluate unlearning performance across three dimensions: *Memorization*, *Utility*, and *Privacy*. All metrics are normalized to the range $[0, 1]$. We denote the unlearned model as $\theta_{unl}$, the original finetuned model as $\theta_{ft}$, and the gold-standard retain model (trained only on $\mathcal{D}_{retain}$) as $\theta_{ret}$.

*Table 6.* Detailed breakdown of unlearning performance on the TOFU dataset (Forget10). The **Overall** score is the harmonic mean of Memorization, Utility, and Privacy scores. **Memorization**, **Utility**, and **Privacy** scores are harmonic means of their respective sub-metrics (denoted in the columns below them). ↑ indicates higher is better, and ↓ indicates lower is better. Best values (excluding baselines) are **bolded**.

| METHOD | OVERALL ↑ | MEMORIZATION | | | | | UTILITY | | | PRIVACY | | | | | FQ ↑ |
|---|---|---|---|---|---|---|---|---|---|---|---|---|---|---|---|
| | | SCORE ↑ | EM ↓ | ES ↓ | PP ↓ | TR ↓ | SCORE ↑ | MU ↑ | FF ↑ | SCORE ↑ | $S_{\text{Loss}}$ ↑ | $S_{\text{Zlib}}$ ↑ | $S_{\text{Min-k}}$ ↑ | $S_{\text{Min-k++}}$ ↑ | |
| INIT FINE. | 0.0000 | 0.0000 | 1.0000 | 1.0000 | 1.0000 | 1.0000 | 1.0000 | 1.0000 | 1.0000 | 0.0038 | 0.0071 | 0.0023 | 0.0062 | 0.0031 | -21.4083 |
| RETAIN | 0.9977 | 1.0000 | 0.0000 | 0.0000 | 0.0000 | 0.0000 | 0.9933 | 0.9866 | 1.0000 | 1.0000 | 1.0000 | 1.0000 | 1.0000 | 1.0000 | 0.0000 |
| GRADDIFF | 0.2270 | 0.1849 | 0.7182 | 0.3358 | 0.5878 | 0.9294 | **0.9934** | 0.9870 | 1.0000 | 0.1471 | 0.1191 | 0.1169 | 0.1221 | 0.4869 | -15.8262 |
| IDKNLL | 0.1456 | 0.4547 | 0.6211 | 0.2715 | 0.5392 | 0.6175 | 0.9102 | 0.8351 | 1.0000 | 0.0578 | 0.0508 | 0.0674 | 0.0476 | 0.0730 | -13.4845 |
| UNDIAL | 0.4129 | 0.5644 | 0.7399 | 0.0340 | 0.0334 | 0.1473 | 0.9155 | 0.8571 | 0.9824 | 0.2272 | 0.6223 | 0.6885 | 0.2008 | 0.1045 | -11.3335 |
| RMU | 0.7568 | 0.8660 | 0.2953 | 0.0108 | 0.0373 | 0.1305 | 0.7471 | 0.8526 | 0.6648 | 0.6799 | 0.6345 | 0.6430 | 0.7150 | 0.7392 | -0.2357 |
| ALTPO | **0.9549** | 0.9254 | 0.0953 | **0.0039** | **0.0095** | 0.1692 | 0.9864 | **0.9836** | 0.9891 | **0.9549** | 0.9380 | **0.9666** | 0.9468 | 0.9688 | -0.7424 |
| NPO | 0.9441 | 0.9339 | 0.0948 | 0.0376 | 0.0147 | 0.1106 | 0.9501 | 0.9049 | 1.0000 | 0.9484 | 0.9231 | 0.9480 | 0.9399 | **0.9846** | -1.9986 |
| SIMNPO | 0.9454 | 0.9435 | 0.1035 | 0.0114 | 0.0359 | 0.0700 | 0.9706 | 0.9429 | 1.0000 | 0.9232 | **0.9882** | 0.9586 | **0.9593** | 0.8095 | -0.0897 |
| IDKDPO | 0.8890 | 0.9532 | 0.1155 | 0.0239 | 0.0249 | 0.0156 | 0.9660 | 0.9343 | 1.0000 | 0.7751 | 0.7529 | 0.8257 | 0.7561 | 0.7700 | -1.1856 |
| ENGRAM ($\bar{\alpha}$) | 0.6984 | 0.9176 | **0.0069** | 0.0071 | 0.0615 | 0.2185 | 0.8801 | 0.7858 | 1.0000 | 0.4832 | 0.6505 | 0.6898 | 0.5618 | 0.2848 | -0.6125 |
| ENGRAM ($\tilde{\alpha}$) | 0.8177 | **0.9627** | 0.0276 | 0.0201 | 0.0842 | **0.0138** | 0.9256 | 0.8616 | 1.0000 | 0.6453 | 0.7688 | 0.7255 | 0.6905 | 0.4829 | **-0.0637** |

### E.2.1. HARMONIC MEAN AGGREGATION

To ensure a balanced evaluation across varying scales, we employ the `safe_hmean` as defined in the OpenUnlearning analysis suite. For a set of scores $S = \{s_1, \ldots, s_n\}$, the harmonic mean is calculated as:

$$H(S) = \frac{n}{\sum_{i=1}^{n} \frac{1}{\max(s_i, \epsilon)}} \quad (26)$$

where $\epsilon = 10^{-10}$ is a small constant to prevent division by zero.

### E.2.2. MEMORIZATION METRICS

These metrics quantify the residual knowledge of $\mathcal{D}_{forget}$. Lower raw values indicate better unlearning, but for the aggregate score, we normalize and invert them so that ↑ indicates better forgetting.

**Exact Memorization (EM)** EM measures the proportion of tokens in the generated response that exactly match the ground truth. We rescale the raw EM score based on the performance gap between the retain and finetuned models. Let $E(\theta)$ be the raw exact match rate of model $\theta$. The rescaled metric is:

$$\text{EM}_{rescaled} = \text{clip}\left(\frac{|E(\theta_{unl}) - E(\theta_{ret})|}{|E(\theta_{ft}) - E(\theta_{ret})|}, 0, 1\right) \quad (27)$$

A value of 0 implies the model behaves exactly like the retain model (ideal unlearning).

**Forget Quality (FQ)** FQ assesses the distributional similarity of model outputs. Following the TOFU protocol ([Maini et al., 2024](#)), we compute the Kolmogorov-Smirnov (KS) test p-value between the Truth Ratio distributions of the unlearned model and the retain model on $\mathcal{D}_{forget}$.

$$\text{FQ}_{raw} = \text{KS\_Test}\left(\text{TR}(\theta_{unl}, \mathcal{D}_{forget}), \text{TR}(\theta_{ret}, \mathcal{D}_{forget})\right) \quad (28)$$

For reporting, we use the logarithmic scale: $\text{FQ} = \log_{10}(\text{FQ}_{raw})$. Higher values (closer to 0) indicate that the unlearned model's probability distribution is statistically indistinguishable from the retain model.

### E.2.3. UTILITY METRICS

These metrics ensure the model preserves capabilities on $\mathcal{D}_{retain}$ and general tasks.

**Model Utility (MU)** MU measures the accuracy on the retain set and real-world knowledge. It is normalized by the performance of the original finetuned model. Let $A(\theta)$ be the aggregate accuracy/utility score of model $\theta$.

$$\text{MU} = \text{clip}\left(\frac{A(\theta_{unl})}{A(\theta_{ft})}, 0, 1\right) \quad (29)$$

**Forget Fluency (FF)** FF ensures the model generates grammatically correct responses even when refusing to answer forget-set queries, rather than generating gibberish. It is calculated using a binary classifier $C_{fl}$ trained to detect linguistic fluency.

### E.2.4. PRIVACY METRICS (INDISTINGUISHABILITY)

We utilize Membership Inference Attacks (MIA) to measure privacy. We employ four distinct MIA methods: **Loss**, **Zlib**, **Min-K**, and **Min-K++**. For each method, we compute the Area Under the Curve (AUC) by distinguishing between the unlearned model's scores on forget samples versus the retain model's scores.

The *Privacy Score* (Indistinguishability) is derived from the AUC. An AUC of 0.5 implies perfect indistinguishability

(ideal privacy).

$$\text{Privacy Score} = 1 - 2 \cdot |\text{AUC} - 0.5| \qquad (30)$$

The final Privacy metric is the harmonic mean of the scores from all four MIA methods.

### E.3. Method Details: Engram

We evaluate two variants of our proposed method, **Engram**, which leverages the norm of weight updates to guide forgetting.

- **Engram ($\alpha = k$):** This variant searches for a best hyperparameter $\alpha$ across all layers. In our experiments, $k = 0.6$ was chosen from search space $\{0.05, 0.10, ..., 0.95, 1.0\}$ (step 0.05).

- **Engram ($\alpha_{\text{W-Norm}}$):** This variant employs an adaptive $\alpha$. We compute the ratio of the weight update norms $\|W^+\|/\|W\|$ and rescale this ratio to the range $[0, 1]$ based on its maximum value across layers. This allows the model to apply stronger unlearning rates to parameters that changed significantly during the initial learning phase.

### E.4. Computational Cost Breakdown

This appendix details the compute and memory profile underlying the compute–accuracy frontier discussion in Section 7. All numbers are computed for Llama-3.2-1B on the TOFU forget10 split ($\sim$ 1M tokens), with batch size 2 and sequence length 256 on a single A100 80GB GPU.

**FLOPs.** For a Transformer with $P$ trainable parameters, the forward pass costs approximately $2P$ FLOPs per token (one multiply–accumulate per parameter), and the backward pass costs approximately $4P$ FLOPs per token (gradients with respect to weights and inputs). For Llama-3.2-1B, $P = 1.24 \times 10^9$, and the TOFU forget10 split contains $N \approx 10^6$ tokens. Standard gradient-based unlearning trains for 10 epochs, while Engram requires a single forward pass to accumulate covariance statistics, followed by a closed-form pseudoinverse solve whose cost is dominated by SVD on each layer's $d \times d$ covariance matrix ($\mathcal{O}(d^3)$ per layer, $\sim$ 0.01 PFLOPs total).

**Memory.** Gradient-based methods store the model weights ($4P$ bytes in FP32), gradients ($4P$), optimizer states (Adam: $8P$), and activations ($\sim 4P$ at our batch/sequence setting). Engram stores only the weights and the layer-wise covariance matrices ($\sim$ 5.92 GB for all linear projections combined).

**Summary.** Table 7 consolidates the breakdown. Engram requires roughly $\sim 30\times$ fewer FLOPs and $\sim 2.3\times$ less peak memory than gradient-based unlearning, with wall-clock times of $\sim$ 2 minutes versus tens of minutes to an hour. This asymmetry is what places Engram on a distinct point of the compute–accuracy frontier discussed in Section 7.

*Table 7.* **Compute and memory profile for gradient-based unlearning vs. Engram** on Llama-3.2-1B (TOFU forget10). Gradient-based estimates assume 10 epochs of training (e.g., AltPO/NPO/SimNPO); Engram requires a single forward pass plus a closed-form solve. $P = 1.24 \times 10^9$ parameters; activations estimated at batch size 2, sequence length 256.

| Resource | Gradient-based | Engram | Ratio |
|---|---|---|---|
| Forward FLOPs | 24.8 PFLOPs | 2.48 PFLOPs | $10\times$ |
| Backward FLOPs | 49.6 PFLOPs | — | — |
| SVD overhead | — | $\sim$ 0.01 PFLOPs | negl. |
| **Total FLOPs** | **74.4 PFLOPs** | **2.5 PFLOPs** | $\sim \mathbf{30\times}$ |
| Weights ($4P$) | 4.96 GB | 4.96 GB | — |
| Gradients ($4P$) | 4.96 GB | — | — |
| Optimizer ($8P$) | 9.92 GB | — | — |
| Activations ($\sim 4P$) | $\sim$ 4.96 GB | — | — |
| Covariance | — | 5.92 GB | — |
| **Total Memory** | $\sim$ **24.8 GB** | $\sim$ **10.9 GB** | $\sim \mathbf{2.3\times}$ |
| Wall time (A100) | $\sim$ 10–60 min | $\sim$ 2 min | $\sim$ 5–30$\times$ |

### E.5. Closed-Form Editing Baselines: Detailed Comparison

This appendix details the implementation of the closed-form editing baselines compared in Table 4 of the main text, along with the hyperparameter sweeps used for fair comparison.

**Algebraic correspondence.** Both UCE (Gandikota et al., 2024) and Engram can be written as

$$W^+ = W \cdot \Sigma^+ \cdot D, \qquad (31)$$

differing only in their regularization choice for $D$:

$$\text{UCE:} \quad D = (\Sigma^+ + \Sigma^- + \lambda I)^{-1}$$
$$\text{Engram:} \quad D = (\Sigma^+ + \Sigma^-)^\dagger$$

UCE's uniform damping $\lambda I$ stabilizes the inverse by perturbing *all* spectral directions, including well-conditioned ones; when these per-layer perturbations accumulate across the full network, they shift output behavior even where the original covariance is well-conditioned. The Moore–Penrose pseudoinverse instead applies a hard spectral cutoff, discarding only the ill-conditioned directions and leaving the well-conditioned subspace untouched. We hypothesize this distinction is the primary source of the observed performance gap.

Task Arithmetic (Ilharco et al., 2023) adopts a different form,

$$W_{\text{new}} = W - \alpha W_{\text{tv}},$$

where the task vector $W_{tv} = W_{ft\ on\ forget} - W_{pt}$ is obtained by fine-tuning the pretrained model on the forget set and subtracting the pretrained weights. Although not strictly a covariance-based estimator, it shares the gradient-free, single-shot paradigm and serves as a natural baseline for closed-form unlearning.

**Hyperparameter sweeps.** For each baseline we report the configuration achieving the best Overall score:

- **Task Arithmetic**: $\alpha \in \{0.5, 1.0, 2.0, 5.0, 20.0\}$ (5 values).

- **UCE**: $\lambda \in \{0, 0.01, 0.05, 0.1, 0.3, 0.5, 1.0\}$ (7 values).

All methods are applied to the same set of Llama-3.2-1B projection layers (Q, K, V, O, Gate, Up, Down across all 16 transformer blocks plus the LM head).

### E.6. Layer-Type Ablation: Empirical Validation of W-Norm Localization

The W-Norm heatmap (Fig. 9) suggests that memory traces concentrate within Query, Key, and MLP-Gate projections in Llama-3.2-1B. To empirically validate this localization, we conduct a layer-type ablation: applying engram extraction selectively to subsets of projection layers and measuring the resulting unlearning performance on TOFU forget10.

*Table 8.* **Layer-type ablation on TOFU** (Llama-3.2-1B, forget10). We selectively extract engrams from different projection layer subsets to isolate the contribution of each layer type. Q/K + Gate alone matches the all-layer Overall score, while removing Q/K/Gate collapses unlearning performance to near-baseline, empirically confirming the W-Norm localization pattern (Fig. 9). The Utility column shows that excluding Q/K/Gate preserves general capability (0.968) at the cost of completely failing to unlearn—indicating that these layers are precisely where the target memory resides.

| Targeted Layers | Overall ↑ | Mem. ↑ | Util. ↑ | Priv. ↑ | EM ↓ |
|---|---|---|---|---|---|
| All linear projections | 0.819 | 0.963 | 0.926 | 0.645 | **0.028** |
| Q/K only | 0.510 | 0.715 | 0.590 | 0.359 | 0.554 |
| **Q/K + Gate** | **0.819** | 0.930 | **0.928** | **0.661** | 0.076 |
| No Q/K/Gate | 0.238 | 0.663 | 0.968 | 0.099 | 0.505 |

**Findings.** Three observations emerge from Table 8. *First*, Q/K + Gate alone achieves Overall 0.819, matching the all-layer configuration (0.819). This confirms that the spatial concentration revealed by the W-Norm heatmap is functionally meaningful: memory traces are not merely *statistically* concentrated in these projections, they are *causally* sufficient for unlearning. *Second*, excluding Q/K/Gate collapses Overall to 0.238 while preserving Utility at 0.968—the resulting model retains general capability but completely fails to unlearn the target, demonstrating that these layers are precisely where the target memory resides. *Third*, Q/K projections alone are insufficient (Overall 0.510); the Gate projection

contributes a critical complementary signal, suggesting that attention-based recall and MLP-based association both participate in storing the unlearned content.

**Implication for scaling.** This finding has practical implications for applying Engram at larger scales. As noted in Appendix H, the dominant storage cost at 8B–70B parameters is the MLP-down covariance ($d = 14336$ for Llama-3 8B). Table 8 suggests that restricting extraction to Q/K + Gate layers eliminates this bottleneck with negligible loss in unlearning quality, making the method viable at scales where storing all-layer covariances would be prohibitive.

## F. Compositional Memory States Hypothesis

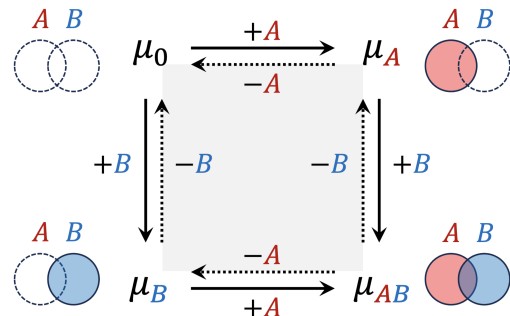

*Figure 17.* **Transitions between fundamental memory states in compositional learning.** For a concept $\mathcal{C}$ consisting of subconcepts $\{A, B\}$, we identify four compositional memory states: (i) the **knowledge vacuum** $\mu_0$; (ii) **isolated states** $\mu_A, \mu_B$ for individual concepts; and (iii) the **composite state** $\mu_{AB}$ for their superposition. Solid and dotted arrows represent acquisition ($+$) and elimination ($-$) processes, respectively. Our framework posits a commutative manifold where the integration of $A$ and $B$ reaches a consistent equilibrium regardless of the learning sequence.

In this section, we provide an intuitive formalization of our framework, grounded in the perspective that learning and unlearning are symmetric operations traversing a structured parameter space.

### F.1. Idealized Parameter States

We proceed with the assumption that for any given concept $c$ (or a combination thereof), there exists an *idealized parameter state* $\theta_c^*$ that optimally represents that knowledge. Under this view, "learning" is the trajectory from an initial state $\theta_0$ to $\theta_c^*$, and "unlearning" is the inverse operation returning the model to a state indistinguishable from one that never observed $c$ (Golatkar et al., 2020; Bourtoule et al., 2021).

Recent studies in *Task Arithmetic* (Ilharco et al., 2023) and *Linear Mode Connectivity* (Frankle et al., 2020) suggest that in the fine-tuning regime, these task-specific updates often reside in a linear subspace. Extending this to our context, we posit that the **Engram** $\mu_c^+$ acts as a task vector connecting these states. Consequently, the transition between memory states can be modeled as vector addition and subtraction in

the weight space:

$$\boldsymbol{\mu}_{\text{learned}} \approx \boldsymbol{\mu}_{\text{before learning}} + \boldsymbol{\mu}_c^+,$$

$$\boldsymbol{\mu}_{\text{after unlearning}} \approx \boldsymbol{\mu}_{\text{learned}} - \boldsymbol{\mu}_c^+.$$

### F.2. Combinatorial States and Zero-shot Traversal

Consider a scenario with $n$ distinct concepts. The set of all possible knowledge combinations (the power set) yields $2^n$ unique memory states, ranging from the **Knowledge Vacuum** (where no concepts are known) to the fully composite state (where all concepts are known).

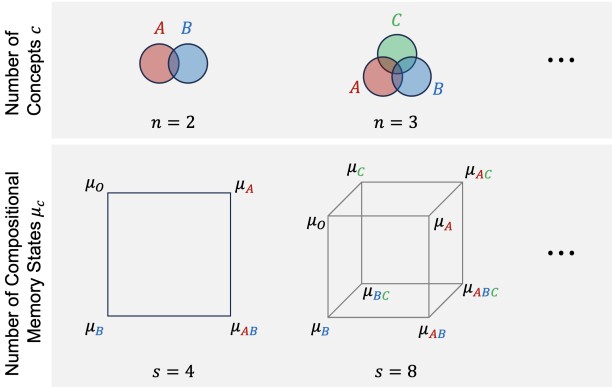

*Figure 18.* **Combinatorial State Space.** For $n$ concepts, the memory manifold consists of $2^n$ discrete nodes. Excluding the trivial starting state (Vacuum), there are $2^n - 1$ active knowledge states that a user might wish to reach (or unlearn to). Our Engrammatic Decomposition allows for zero-shot traversal between any two nodes in this hypercube via simple arithmetic.

As illustrated in Figure 18, if we exclude the trivial starting point, there are $2^n - 1$ **distinct semantic states** that a model might need to adopt. Standard unlearning approaches would require iterative optimization to reach each specific state (e.g., "forget A but keep B"). In contrast, our framework leverages the linear compositionality of engrams to synthesize any of these $2^n - 1$ states instantaneously, effectively resolving the combinatorial complexity of selective forgetting without additional training.

## G. Structural Role of the Tabula Rasa Instantiation

Section 4.5 adopts the instantiation $\Delta W \approx W_{\text{ft}} - \mathbf{0}$ rather than the fine-tuning delta $\Delta W = W_{\text{ft}} - W_{\text{pt}}$. This appendix justifies the choice on three grounds: (i) the two choices yield *structurally different* optimization problems, not merely numerically different estimators; (ii) the tabula rasa form is what makes the estimator single-pass and parallelizable across layers; (iii) the delta-weight form, while attractive interpretively, would require iterative sequential optimization that is impractical at LLM scale. We also report the empirical consequence of substituting the delta naively without such optimization.

### G.1. Three-State Derivation

Consider three reference states $s \in \{0, 1, 2\}$:

- $s = 0$: the null state, $W_0 = \mathbf{0}$;
- $s = 1$: the pretrained state, $W_1$;
- $s = 2$: the fine-tuned state, $W_2$.

The general closed-form editing solution, following the MEMIT/UCE form (Meng et al., 2023; Gandikota et al., 2024), can be written as

$$W_{\text{new}} = \left(V^* X^{+\top} + W_2 X^- X^{-\top}\right) D \qquad (32)$$

where $D = (\Sigma^+ + \Sigma^-)^\dagger$ and $V^*$ denotes the desired pre-activation output at the edited layer. The choice of source/target state determines $V^*$ and, through it, the structure of the resulting optimization.

**Case 1: Tabula rasa** ($s = 2 \to s = 0$). Setting $V^* = W_0 X^* = \mathbf{0}$ in Eq. (32):

$$W_{\text{new}} = \big(\underbrace{W_0 X^*}_{= \mathbf{0}} \cdot X^{+\top} + W_2 X^- X^{-\top}\big) D$$
$$= W_2 \Sigma^- (\Sigma^+ + \Sigma^-)^\dagger.$$

The first term vanishes at *every* layer, *independently of the input*. Each layer's solution depends only on its own weights and activation statistics, with no dependence on the target activations of neighbouring layers. All $L$ layers can therefore be solved in parallel—this is the structural property exploited by Eq. (4) in the main text.

**Case 2: Delta weights** ($s = 2 \to s = 1$). Setting $V^* = W_1 X^*$ in Eq. (32):

$$W_{\text{new}} = \big(\underbrace{W_1 X^*}_{\neq \mathbf{0}} \cdot X^{+\top} + W_2 X^- X^{-\top}\big) D.$$

The first term survives. Crucially, editing layer $\ell$ shifts the activations entering layer $\ell+1$, which in turn shifts the required $V^{*(\ell+1)}$ for the next layer. This creates a *cascading dependency* across the network's depth, so a rigorous solution to Case 2 requires **backward sequential optimization** of $V^*$ from later to earlier layers—structurally analogous to the optimization loop in MEMIT (Meng et al., 2023).

**Summary.** The tabula rasa instantiation is not merely a numerical convenience: it is the structural condition under which $V^*$ collapses to zero, removing the cross-layer coupling that would otherwise force sequential optimization. Single-pass parallelism is a property of Case 1, not of the estimator template itself.

## G.2. Empirical Consequence of Naive Substitution

A natural question is whether the delta-weight form can still be applied *without* performing the sequential $V^*$ optimization required by Case 2—i.e., by simply substituting $\Delta W = W_{\text{ft}} - W_{\text{pt}}$ into Eq. (4). Table 9 reports the result on TOFU (Llama-3.2-1B, forget10). The per-layer error from ignoring the cascading dependency accumulates across the network's depth, collapsing Overall performance ($0.818 \rightarrow 0.446$) and degrading Exact Memorization suppression by more than an order of magnitude ($0.028 \rightarrow 0.617$).

*Table 9.* Effect of the instantiation choice on TOFU (Llama-3.2-1B, forget10). The tabula rasa instantiation decouples the layer-wise sub-problems and solves all layers in parallel. Naive substitution of the fine-tuning delta without sequential $V^*$ optimization accumulates per-layer error.

| Instantiation | Overall ↑ | EM ↓ |
|---|---|---|
| $\Delta W = W_{\text{ft}} - W_{\text{pt}}$ (naive) | 0.446 | 0.617 |
| $\Delta W = W_{\text{ft}}$ (tabula rasa, **ours**) | **0.818** | **0.028** |

## G.3. Why We Do Not Pursue Sequential Optimization

In principle, Case 2 admits a rigorous solution via MEMIT-style sequential editing: solve $V^*$ from the deepest layer backward, re-propagating activations after each layer's edit. While this would yield an estimator that more directly identifies the perturbation contributed by concept $C$ during fine-tuning, the cost is prohibitive at the scales we target. ROME (Meng et al., 2022) and MEMIT typically operate on 1–6 transformer layers, where sequential coordination is tractable. Our framework targets all linear projection layers of Llama-3.2-1B simultaneously ($W_Q, W_K, W_V, W_O$ across 16 attention blocks and $W_{\text{gate}}, W_{\text{up}}, W_{\text{down}}$ across 16 MLP blocks, plus the LM head). Sequential optimization at this scale would require backpropagation through every layer for every concept, eliminating the closed-form efficiency that motivates our framework.

**Interpretation.** The trade-off is interpretive rather than technical: the tabula rasa estimator identifies a *functionally separable component of the converged manifold*—validated by the sufficiency and necessity tests in experimental results—rather than reconstructing the specific weight perturbation that concept $C$ contributed during learning. We retain the engram terminology because the extracted components causally satisfy the four neuroscience criteria, while explicitly acknowledging that the underlying mechanism is functional identification of the converged state.

## H. Limitations

**Functional Identification, Not Learning Trajectory Reconstruction.** Strictly speaking, the components isolated by Eq. (4) are *functionally separable subspaces of the trained manifold*, not reconstructions of the weight perturbations that each concept contributed during learning. The tabula rasa instantiation (Appendix G) makes this identification well-posed and single-pass, at the cost of conflating contributions from pretraining and fine-tuning into a single converged representation. We retain the term "engram" because the extracted components causally satisfy the four neuroscience criteria of specificity, reactivation, sufficiency, and necessity (Section 5), and because they support compositional manipulation through linear arithmetic. We emphasize, however, that the underlying mechanism is the identification of a functional component of the converged state, rather than a reconstruction of the learning trajectory itself—an interpretive distinction that we view as a feature, not a limitation, of closed-form spectral memory isolation.

**Requirement for Concept Datasets.** Unlike unsupervised feature discovery methods such as Sparse Autoencoders (Huben et al., 2024) or Linear Parameter Decomposition (Braun et al., 2025; Bushnaq et al., 2025), our framework requires explicit construction of target ($X^+$) and reference ($X^-$) datasets. This necessitates human labeling or domain knowledge to define concepts, limiting applicability in fully unsupervised discovery scenarios. A promising extension is to combine the estimator with concept discovery methods—e.g., using SAE-identified features to automatically construct $X^+/X^-$ partitions—enabling fully unsupervised engram extraction.

**Retrospective-Only Estimation.** The current framework operates retrospectively on fully trained models. It cannot be integrated into the training loop for online engram tracking or applied in streaming/incremental learning scenarios where real-time memory isolation might be desirable.

**Covariance Matrix Storage at Scale.** Although our estimator reduces space complexity from $\mathcal{O}(Nd)$ to $\mathcal{O}(d^2)$, storing $d \times d$ covariance matrices for each layer can become substantial at 8B–70B parameter scale. A layer with $d = 4096$ requires approximately 67 MB per covariance matrix in single precision, accumulating across layers in deep architectures. Two complementary mitigations are available: (i) truncated SVD storage exploiting the typically low effective rank of these matrices, and (ii) restricting extraction to Q/K/Gate layers identified by W-Norm analysis (Section 7), which achieves comparable Overall performance while eliminating the expensive MLP-down covariance.

