# OpenReview forum: "AI Engram: In Search of Memory Traces in Artificial Intelligence"
_ICML.cc/2026/Conference — ICML 2026 spotlight_

### Official Review · Reviewer_3sNs · 2026-03-07

**Soundness:** 3
**Presentation:** 2
**Significance:** 3
**Originality:** 4
**Overall Recommendation:** 5
**Confidence:** 4

**Summary:**

The paper introduces “AI engrams”, a framework to isolate memory traces associated with specific concepts in trained neural networks. Starting from neuroscientific criteria (specificity, reactivation, sufficiency, necessity), the authors formulate a constrained inverse problem in pre-activation space and derive a closed-form estimator that projects trained weights onto concept-specific subspaces defined by target/reference covariances. They show this estimator coincides with a Fisher-metric projection / natural gradient step, and empirically demonstrate class-wise and attribute-wise unlearning, compositional “engram arithmetic”, and partial success on LLM unlearning benchmarks.

**Compliance With Llm Reviewing Policy:**

Affirmed.

**Final Justification:**

The authors' response successfully resolved my main questions regarding implementation and applicability, leading me to raise my rating. I consider this an impressive work that should be accepted. The method is both novel and practical, and the rebuttal has solidified my trust in the results. My only remaining suggestion concerns the presentation: Section 3 is quite dense and abstract, which may hinder readability. I recommend adding specific examples to clarify the core ideas and improving the general prose to ensure a smoother reading experience.

**Key Questions For Authors:**

1. Can engram be directly generated and inserted into the model instead of only being erased, similar to a mapping from data to parameters? If achievable, this might be highly beneficial for context internalization.
2. For the experiments on Llama-3.2-1B, how did you handle the huge intermediate layer dimensions of the MLP within the Transformer? Did you apply the method to all layers or only specific types of layers (such as Attention Q/K)? (Figure 9 in the paper implies a sensitivity analysis for different layers, but we hope the specific implementation details can be clarified, and it would be best to open-source the code.)

**Limitations:**

yes

**Strengths And Weaknesses:**

Strengths:
1. The form is highly concise with a closed‑form solution. From the application perspective of concept erasure and editing, it eliminates a great deal of iterative optimization and thus has strong engineering appeal.
2. Narratively, it connects neuroscientific engram standards, linear algebraic constraints, and Fisher/natural gradient, allowing readers to understand it from a broader perspective of optimization geometry.
3. The experiments cover a wide range, from MLP/CNN/ViT to LLMs and further to attribute editing in generative models, showing a large experimental span that at least demonstrates transferability.

Weaknesses:
1. The writing needs further improvement; the content is difficult to read, and some abstract concepts are hard to understand.
2. The storage overhead of the covariance matrix is excessive. Although the time complexity is relatively low, the method requires computing and storing a $d \times d$ covariance matrix for each layer.

---

> ### Author Rebuttal · Authors · 2026-03-31
>
> We thank the reviewer for recognizing the engineering appeal and the originality of connecting neuroscientific criteria with optimization geometry.
>
> **Please see new figures/tables:** https://files.catbox.moe/8lokoj.pdf
>
> ---
>
> > **Q1.** Can engrams be directly generated and inserted into the model instead of only being erased? This might be highly beneficial for context internalization.
>
> Yes, we find this direction equally exciting. Our framework already demonstrates insertion through engram arithmetic (Section 5.3, Fig. 8): injecting W⁺ into a model induces the target attribute, validating the *sufficiency* criterion. Encouraged by this, we tested whether the same principle extends to LLMs.
>
> **\[Table R5\]**: Using contrastive covariance from just 20-shot ICL examples on Llama-3.2-1B (Todd et al., ICLR 2024), we apply W' \= W \+ α·W⁺ with no gradient computation. The engram recovers \~60% of ICL performance at zero inference overhead. The reviewer's intuition is exactly right: this positions the engram as a lightweight **data → parameter** mapping. A single forward pass converts a few demonstrations into a persistent weight-space edit, bypassing repeated inference-time context. This opens a promising direction we plan to explore.
>
> ---
>
> > **Q2.** For Llama-3.2-1B, how did you handle the huge MLP dimensions? All layers or specific types? Please open-source the code.
>
> We applied the method to **all linear projection layers**: Q, K, V, O in self-attention and Gate, Up, Down in MLP across all 16 transformer layers simultaneously. The MLP down projection uses a 2.7–4× expansion factor (d \= 8192 in Llama-3.2-1B), so its covariance Σ ∈ ℝ^{8192×8192} incurs \~8–16× more storage than attention projections (d \= 2048). At the 1B scale this is manageable, but we acknowledge that scaling to 8B–70B models would make full d×d storage challenging.
>
> One possible idea for this is to store covariances in a truncated SVD form (d × k factors, k ≪ d), substantially reducing the footprint. Combined with **\[Table R2\]**'s finding — Q/K \+ Gate alone matches all-layer Overall (0.819 vs. 0.818) while removing Q/K/Gate collapses to 0.238 — restricting extraction to these smaller layers could eliminate the MLP bottleneck entirely. Whether this generalizes beyond TOFU is open, but **\[Figure R3\]** shows consistency across forget set sizes. We have prepared the full implementation — extraction scripts, layer-wise pipeline, and reproduction notebooks — and will release the codebase upon acceptance.
>
> ---
>
> > **W1.** The writing needs improvement; some abstract concepts are hard to understand.
>
> We appreciate this feedback and take it seriously. Could the reviewer kindly point us to the specific sections or concepts that were most difficult to follow? This would help us target our revisions precisely. In the meantime, we plan to add intuitive overview paragraphs before formal derivations and include a concrete worked example illustrating the extraction pipeline. We are committed to improving readability in the camera-ready.
>
> ---
>
> > **W2.** The storage overhead of the covariance matrix is excessive.
>
> **\[Table R4\]** contextualizes this: Engram total is \~10.9 GB vs. \~24.8 GB for gradient-based methods at the 1B scale. For larger models (8B–70B), full d×d storage becomes a real bottleneck. As discussed in Q2, two complementary paths can address this: (1) truncated SVD storage (**\[Figure R4\]**'s rtol stability implies effective rank ≪ d), and (2) Q/K/Gate-only extraction (**\[Table R2\]**), which eliminates the expensive MLP covariances with virtually no performance loss. We believe combining these approaches can make the method viable at larger scales.

---

> > ### Author Rebuttal · Reviewer_3sNs · 2026-04-02
> >
> > The author's rebuttal resolved my questions. I think this is a good paper and should be accepted, so I am increasing my rating. That said, Section 3 is a bit of a tough read. I recommend adding some concrete examples to clarify the abstract concepts and improve the overall flow.

---

> > > ### Author Response · Authors · 2026-04-05
> > >
> > > We are grateful for the reviewer’s positive assessment and the score update. The feedback has been helpful for improving our work and we will be sure to add concrete examples in Section 3 as suggested.

---

### Official Review · Reviewer_brom · 2026-03-10

**Soundness:** 3
**Presentation:** 4
**Significance:** 3
**Originality:** 2
**Overall Recommendation:** 5
**Confidence:** 3

**Summary:**

The paper introduces a framework for identifying concept-specific “AI engrams” in trained neural networks. Starting from neuroscience-inspired criteria such as specificity, sufficiency, necessity, and reactivation, the authors formulate engram identification as a constrained inverse problem in pre-activation space and derive a closed-form layerwise estimator based on target and reference covariances. The framework is then used for class-wise unlearning, semantic editing, combinatorial memory arithmetic, and LLM unlearning, with an additional Fisher/K-FAC interpretation that links the estimator to a projection under information geometry.

**Compliance With Llm Reviewing Policy:**

Affirmed.

**Final Justification:**

I raised my score after the rebuttal because the additional experiments and clarifications substantially addressed the concerns I had raised about robustness to target/reference choices and the intended scope of the method.  Overall, the rebuttal resolved the issues that mattered most to my evaluation.

**Key Questions For Authors:**

please see weakness.

**Limitations:**

yes

**Strengths And Weaknesses:**

Strengths:
- The paper has a coherent mathematical core built around Equations (2)-(4), and Table 1 gives a clear formulation of linear-algebraic constraints.
- The same estimator is used throughout the paper, which keeps the narrative coherent across many experiments.
- Translating neuroscience-inspired engram criteria into a closed-form editing framework is original and conceptually interesting.
- The Fisher/K-FAC interpretation adds a useful theoretical perspective rather than presenting the estimator as a purely heuristic trick.

Weaknesses:
- The framework depends critically on how the target and reference datasets are defined, but robustness to this choice is not studied deeply.
- The framework is most directly applicable when concept datasets can be explicitly constructed, which may limit its use in less supervised or more open-ended settings.

---

> ### Author Rebuttal · Authors · 2026-03-31
>
> We sincerely thank the reviewer for positive assessment and for recognizing the mathematical core and the Fisher/K-FAC interpretation.
>
> **Please see new figures/tables:** https://files.catbox.moe/8lokoj.pdf
>
> ---
>
> > **W1.** The framework depends critically on how target and reference datasets are defined, but robustness to this choice is not studied deeply.
>
> We appreciate this important feedback. We address this from two angles.
>
> First, **imprecise concept boundaries**: **\[Figure R1\]** (see also response to Reviewer fa2V) shows that even when the target shares significant feature overlap with reference classes (same CIFAR-100 superclass), the estimator degrades smoothly (0.80%p drop) and proportionally — not catastrophically. Minor errors in what constitutes "target" vs. "reference" do not cause failure.
>
> Second, **sample quantity**: **\[Figure R2\]** varies the forget set from 5% to 100% of available samples. The retain accuracy is completely unaffected across the entire range on both CIFAR-10 and CIFAR-100. Forget accuracy decreases monotonically with more samples, reflecting progressively more precise estimation of the concept's input subspace. We conjecture that stable estimation requires the number of samples to exceed effective rank and once the principal subspace is fully identified, additional data provides diminishing returns. We plan to formalize this connection in the camera-ready.
>
> ---
>
> > **W2.** The framework is most directly applicable when concept datasets can be explicitly constructed, limiting use in less supervised or open-ended settings.
>
> We agree this is an important consideration. However, explicitly specifying which data to forget is a shared prerequisite of all unlearning methods evaluated in this work — our framework does not impose additional requirements beyond this, and in fact requires *less*, as it needs only a single forward pass rather than iterative optimization.
>
> Extending to unsupervised settings — e.g., combining the estimator with SAE-identified features to automatically construct target/reference partitions — is a promising direction we plan to explore.

---

> > ### Author Rebuttal · Reviewer_brom · 2026-04-04
> >
> > I thank the authors for their rebuttal. I appreciate the additional experiments, which substantially address my concerns. I will raise my score accordingly.

---

> > > ### Author Response · Authors · 2026-04-05
> > >
> > > We sincerely thank the reviewer for the engaging and productive discussion. We will be sure to update our manuscript based on these insights. If the reviewer feels all points have been sufficiently addressed, we would be so grateful if the final evaluation could be updated to reflect this!
> > >
> > > We remain available for any further clarification that may need.

---

### Official Review · Reviewer_fa2V · 2026-03-12

**Soundness:** 2
**Presentation:** 3
**Significance:** 2
**Originality:** 3
**Overall Recommendation:** 4
**Confidence:** 3

**Summary:**

The paper frames memory isolation in neural networks as a constrained inverse problem derived from neuroscience engram criteria (specificity, reactivation, sufficiency, necessity). A closed-form spectral estimator projects trained weights onto concept-specific subspaces using input covariance statistics. The authors claim a connection to natural gradient via K-FAC and demonstrate unlearning across MLPs, CNNs, ViTs, and Llama-3.2-1B.

**Compliance With Llm Reviewing Policy:**

Affirmed.

**Final Justification:**

The core contribution is original and technically sound: a closed-form spectral estimator for memory isolation derived from neuroscientific criteria, with a clean Fisher/K-FAC connection. The rebuttal addressed most of my concerns. Figure R1 (graceful degradation under semantic overlap) satisfies my stated conditional. Table R1 (UCE/TA comparisons) and Table R2 (Q/K/Gate ablation) fill key empirical gaps. I raised my score from 3 to 4.

Two issues remain. Table R3 shows the method requires the full weight matrix rather than the fine-tuning delta, which undermines the biological "engram" framing. The camera-ready should reframe what the estimator actually computes. The LLM results still trail dedicated methods on Overall/Privacy, though the efficiency advantage is real. Neither concern blocks acceptance, but both need honest treatment in revision.

**Key Questions For Authors:**

1. How does performance degrade when forget/retain classes share significant feature overlap (e.g., cat vs. dog, or semantically similar CIFAR-100 superclasses)? If the null-space assumption fails gracefully this would raise my score; if it fails catastrophically, that's a serious problem.
2. Can you conduct UCE and Task Arithmetic comparisons? These seem like the most natural baselines. If Engram doesn't outperform them, the contribution narrows considerably.
3. For Llama, did you try restricting engram extraction to only Q/K/Gate layers per your own W-Norm analysis? Seems like the obvious experiment and its absence is puzzling.
4. How sensitive is the pseudoinverse to the singular value threshold (epsilon = 1e-6 * lambda_max)? If results shift meaningfully with this choice, the "hyperparameter-free" framing is misleading.

**Limitations:**

Appendix G covers some limitations but avoids the hardest ones. The orthogonality assumption's failure mode for related concepts is not discussed. The gap between what "engram" implies (causal memory unit) and what's computed (covariance projection) deserves more honest treatment. Societal impact discussion is cursory.

**Strengths And Weaknesses:**

### Strengths

- The constrained optimization formulation (Eq. 2-4) is mathematically clean and the derivation from biological criteria to a projection operator is well-executed. The layer-wise decomposition into covariance-based projections is practical.
- O(d^2) complexity with single-pass covariance accumulation is genuinely useful. The ImageNet-1k ViT experiment (Fig. 12) shows the method doesn't collapse at scale, which is worth something.
- The engram weight norm heatmap on Llama (Fig. 9) showing concentration in Q/K and MLP gate layers is an interesting empirical observation, though I'm not sure how robust this finding is across different forget sets.
- Combinatorial unlearning via linear arithmetic (Section 4.6) is a nice property on paper.

### Weaknesses

- The tabula rasa assumption (Delta W = W - 0) is doing enormous work and I don't think the paper grapples with this fully. For pretrained models like ViT and Llama, the "initial state" encodes massive prior knowledge. You're not extracting "what concept C contributed," you're applying a covariance-weighted projection to the full weight matrix. Calling this an "engram" oversells what's actually being computed. The biological analogy breaks down precisely where it matters most.
- LLM results are weak. On TOFU (Table 3), Engram trails AltPO, NPO, SimNPO on the overall metric, and the privacy scores are poor (0.48 and 0.65 vs. 0.95 for AltPO). The paper highlights exact memorization suppression but this is cherry-picking. If your method can't compete on standard benchmarks, the theoretical elegance doesn't buy you much.
- The hard null-space constraint W+X- = 0 assumes concepts occupy orthogonal subspaces. This is almost certainly false in any realistic network. The paper acknowledges this in Remark A.2 but then proceeds as if it doesnt matter. I'd expect this to fail badly for semantically overlapping categories, and no experiment tests this.
- The Fisher-Engram "equivalence" (Theorem 6.1) requires isotropic output curvature G_l = sigma^2 I, which throws away all task-specific information. What remains is just a statement about input covariance geometry. The theorem is technically correct but the assumptions are so strong that the connection to actual natural gradient optimization is tenuous at best.
- No comparison against UCE or Task Arithmetic, which are the two most obvious baselines given the paper's own framing. The vision unlearning comparisons also lack recent strong methods. Without these, I can't assess whether the engram estimator offers anything beyond existing closed-form editing approaches.

### Minor

- Fig. 14 caption is clearly wrong (copy-pasted from Fig. 13, describes CIFAR-100/ResNet-18 instead of CelebA).
- The "causal" language throughout is generous. Projecting weights by input covariance is correlational. The sufficiency/necessity tests in Table 1 are closer to causal, but the estimator itself doesn't require interventional data.

---

> ### Author Rebuttal · Authors · 2026-03-31
>
> We thank the reviewer for the rigorous and detailed critique. It identified genuine gaps that led to substantial new experiments.
>
> **Please see new figures/tables:** https://files.catbox.moe/8lokoj.pdf
>
> ---
>
> > **Q1/W3.** How does performance degrade under semantic overlap? "If the null-space assumption fails gracefully this would raise my score."
>
> **[Figure R1]** on CIFAR-100/ResNet-18: (a) same-superclass pairs (high overlap) show a measurable 0.80%p mean accuracy drop, while different-superclass pairs shows near zero, indicating the estimator is sensitive to semantic proximity but degrades proportionally; (b) within a superclass, degradation vs. cosine distance is smooth and monotonic, indicating no catastrophic collapse.
>
> This is because W⁺X⁻ = 0 is the optimization's starting point, not the output. We discuss the underlying mechanism in Section 6.3 ("Near-orthogonality from Lagrangian structure"): P⁺ = Σ⁺(Σ⁺ + Σ⁻)† is a soft projection (P² ≠ P) and shared covariance directions receive attenuated weight, not binary exclusion. Fig. 4 confirms this: cat↔dog and automobile↔truck show visible cross-interference. We will add Figure R1 to the appendix and clarify that W⁺X⁻ \= 0 is a *requirement not the final output*.
>
> ---
>
> > **Q2/W5-1.** UCE and Task Arithmetic comparisons?
>
> **[Table R1]** on TOFU: Engram (Overall 0.818) outperforms UCE (0.659) by \+0.159 and Task Arithmetic (0.584) by +0.234, with **5× lower EM** (0.028 vs. 0.135).
>
> Both UCE and ours share the form $W^{+} \= W \\cdot \\Sigma^{+} \\cdot D$, differing in regularization (UCE: $\\lambda I$ damping, Engram: pseudoinverse cutoff). The pseudoinverse avoids cumulative per-layer perturbations that uniform damping introduces. We will clarify this point.
>
> ---
>
> > **Q4.** Pseudoinverse sensitivity to threshold?
>
> We agree rtol is a hyperparameter, though **[Figure R4]** shows it is stable across a broad range (1e-8 to 1e-3) and the default works without tuning. Will clarify this in the appendix.
>
> ---
>
> > **W2/W5-2.** LLM results trail AltPO/NPO/SimNPO. EM is cherry-picking. Privacy poor. Vision baselines lack recent methods.
>
> As we aim to identify memory traces, unlearning is one of the key validation modalities. We prioritized *architectural breadth* and *computational efficiency* (**[Table R4/R5]**: fewer FLOPs/Storage vs. grad methods).
>
> Regarding the EM results: we report the best Overall (harmonic mean of Mem., Util. & Priv.) configuration and EM was not separately optimized. **[Figure R5]** visualizes the full hyperparameter search landscape (Overall vs. EM). On privacy: this metric is often omitted in LLM unlearning benchmarks, but we included it for transparency and to show that adaptive α (W-Norm) yields broad improvement across all axes including Privacy. Section 7 describes Overall as "*constrained*" and we agree this gap exists relative to dedicated unlearning algorithms, possibly due to the lack of iterative refinement process with backward information.
>
> ---
>
> > **Q3/S3.** Restricting to Q/K/Gate? Is the W-Norm finding robust across forget sets?
>
> **[Table R2]**: Q/K \+ Gate matches all-layer Overall (0.819 vs. 0.818); removing Q/K/Gate collapses to 0.238. **[Figure R3]** confirms this robust pattern across forget01 and forget05. This finding is  informative for memory localization.
>
> ---
>
> > **W1.** Tabula rasa does enormous work. You're not extracting "what concept C contributed."
>
> **[Table R3]**: ΔW \= W\_ft yields Overall **2× higher** than W\_ft − W\_pt (0.818 vs. 0.446). The intuition is that if the pretrained model already learned "cat" from ImageNet-21k, fine-tuning on CIFAR-10 adds only a subtle delta. Thus projector needs the full converged manifold, not the thin fine-tuning residual.
>
> ---
>
> > **W4.** Fisher-Engram requires isotropic G\_l \= σ²I, too strong.
>
> Thanks for the suggestion and we agree this assumption can be relaxed. Its value is showing that two independent paths (biological criteria, information geometry) converge to the same estimator, to provide structural insight, not operational requirement. Empirical results hold regardless. We will reframe Theorem 6.1 accordingly.
>
> ---
>
> > **Limitations.** The gap between what "engram" implies and what's computed deserves more honest treatment.
>
> This is important feedback. We agree our closed-form solution is a first-order spectral approximation of the functional engram. We used the term Engram because the identified subspaces consistently pass the neuroscientific “gold-standard” tests (sufficiency and necessary in Section 5.1) and demonstrated the isolated units are functionally causal. We will add a detailed societal impact discussion including this as well as the potential misuse of engram isolation.
>
> ---
>
> > **Minor.** Fig. 14: copy-paste error. On "causal" language.
>
> On "causal" language: agreed. The estimator is correlational, and causal claims should only apply to validation. We will distinguish consistently. We confirm this is an error and will update as CelebA WAE. Thank you!

---

> > ### Author Rebuttal · Reviewer_fa2V · 2026-04-03
> >
> > The new experiments on semantic overlap (Figure R1), UCE/Task Arithmetic baselines (Table R1), Q/K/Gate ablation (Table R2), and rtol sensitivity (Figure R4) are responsive and resolve the majority of my concerns. The graceful degradation result satisfies my stated conditional, and I am raising my score from 3 to 4.
> >
> > My remaining concern is the tabula rasa issue (W1), which I believe Table R3 sharpens rather than resolves. If ΔW = W_ft works and ΔW = W_ft − W_pt does not, the estimator depends on the full converged weight matrix, not on what a specific concept contributed during learning. This means the method extracts a covariance-weighted slice of the entire trained representation, which is not what "engram" implies biologically.
> >
> > Can the authors clarify how they reconcile this with the engram framing? Specifically: would they be willing to reframe the contribution in the camera-ready as identifying a functionally separable component of the trained manifold, rather than a memory trace of concept-specific learning?
> >
> > A secondary point: the LLM privacy/overall gap relative to AltPO/NPO/SimNPO is acknowledged but not closed. I would recommend framing this as an explicit compute-accuracy tradeoff in the revision.

---

> > > ### Author Response · Authors · 2026-04-05
> > >
> > > We sincerely thank the reviewer for thoughtful encouragement and raising the score. We are also glad to have the opportunity to respond to the reviewer’s follow-up comments on tabula rasa, which helps sharpen the conceptual clarity of our work.
> > >
> > > ---
> > >
> > > > Table R3 sharpens the issue: if W\_ft works but W\_ft − W\_pt does not, the estimator depends on the full converged weight matrix, not on what a concept contributed during learning. Would you reframe as identifying a functionally separable component rather than a memory trace?
> > >
> > > **On the delta formulation.** First, we would like to address why the delta formulation underperforms in **\[Table R3\]**. Reverting a model from its fine-tuned state to a pre-trained state is not as simple as substituting $\\Delta W \= W\_2 \- W\_1$ in **\[Table R3\]** into our estimator. Thanks to the reviewer’s comment, we have reached the a more nuanced understanding of this process, which we wish to add in the revised manuscript.
> > >
> > > Consider three conditions:
> > >
> > > - $s=0$ ($W\_0 \= 0$, null state).
> > > - $s=1$ ($W\_1$, pretrained state).
> > > - $s=2$ ($W\_2$, fine-tuned state).
> > >
> > > Consider the general closed-form editing solution (cf. MEMIT/UCE):
> > >
> > > $$W\_{\\text{new}} \= \\left(V^{\*} X^{+\\top} \+ W\_2 X^{-} X^{-\\top}\\right) D$$
> > >
> > > where $D \= \\left(\\Sigma^{+} \+ \\Sigma^{-}\\right)^{\\dagger}$ and $V^{\*}$ is the desired pre-activation output.
> > >
> > > **Case 1: Tabula rasa ($s=2 \\to s=0$).** Setting $V^\* \= W\_0 X^{\*}$:
> > >
> > > $$W\_{\\text{new}} \= \\left(\\underbrace{W\_0 X^{\*}}\_{=0} \\cdot X^{+\\top} \+ W\_2 X^{-} X^{-\\top}\\right) D$$
> > >
> > > **Case 2: Delta weights ($s=2 \\to s=1$).** Setting $V^\* \= W\_1 X^{\*}$:
> > >
> > > $$W\_{\\text{new}} \= \\left(\\underbrace{W\_1 X^{\*}}\_{\\neq0} \\cdot X^{+\\top} \+ W\_2 X^{-} X^{-\\top}\\right) D$$
> > >
> > > In Case 1, the null state makes the first term to vanish at every layer, independent of the input. Because no optimization is required, every layer can be processed in parallel; this is the core of our engram estimator.
> > >
> > > Case 2 is fundamentally different. The first term survives, and editing layer $\\ell$ changes the activations received by layer $\\ell+1$. This, in turn, changes the required $V^{\*}$, creating a cascading dependency.
> > >
> > > A rigorous solution for Case 2 requires backward sequential optimization of $V^{\*}$ from later to earlier layers (as in MEMIT). Table R3 applied a simplified approach of substituting the delta directly, which accumulates per-layer optimization errors across the model’s depth. While iterative optimization would likely yield more selective engrams as reviewer mentioned, it requires substantial backpropagation and sequential layer coordination. Such computation becomes computationally impractical when targeting all 112 layers of Llama3-1B, compared to 1-6 layer targeting for ROME or MEMIT.
> > >
> > > **The role of tabula rasa.** Tabula rasa is indeed doing significant work, but as a structural necessity rather than a simplification. By collapsing $V^{\*}$ to zero, every layer becomes self-contained, requiring only the current weights and its own activation statistics. This eliminates the need for backward passes, sequential coordination, or iterative optimization.
> > >
> > > Tabula rasa is the structural condition that makes our estimator genuinely **single-pass across all layers.** This is the fundamental distinction that separates our method from the MEMIT family. We will formalize this insight as a remark in the revised manuscript and discuss the theoretical justification for tabula rasa that was previously missing.
> > >
> > > **On reframing.** We agree with the reviewer's distinction: AI engrams identify functionally separable components of the trained manifold. The tabula rasa condition makes this identification well-posed and single-pass. While we wish to retain the engram terminology because the extracted components satisfy the neuroscience criteria (sufficiency, necessity, specificity, reactivation), we will mention in our manuscript that the underlying mechanism is functional identification of the converged state, not a reconstruction of the learning trajectory.
> > >
> > > We sincerely appreciate the reviewer’s insightful feedback on tabula rasa that helped us further sharpen the conceptual clarity.
> > >
> > > ---
> > >
> > > > A secondary point: the LLM privacy/overall gap relative to AltPO/NPO/SimNPO is acknowledged but not closed. I would recommend framing this as an explicit compute-accuracy tradeoff in the revision.
> > >
> > > We agree to this point and will frame it into compute-accuracy tradeoff explicitly in the revision with Table R4/R5. Thank you.

---

### Official Review · Reviewer_PEYZ · 2026-03-13

**Soundness:** 3
**Presentation:** 3
**Significance:** 3
**Originality:** 3
**Overall Recommendation:** 4
**Confidence:** 4

**Summary:**

This paper introduce a neurosceinfically inspired framework for isolation of memory units (AI engrams) in deep neural netwroks with proper analytical support. A key advantage the method is the ability to reconstruct the underlying memory engrams of each concept in one forward pass.
As a proof of concept the authors also show the composability of these reconstructed engrams to generate new concepts, with robust validations across concepts, architectures, and tasks.

The theoretical practices in this paper are well structured and help with the clarification of intuition behind the engram design.  Moreover the  theoretical lens supports how the biological inspired method lead to natural gradient update on the parameter manifold, providing geometric explanation for the empirical results in the paper.

This paper is a meaningful contribution to NeuroInspired AI literature, which will be of interest to both neuroscience and AI community, regardless of publication in this venue.

**Compliance With Llm Reviewing Policy:**

Affirmed.

**Key Questions For Authors:**

1) For equation 4, you assumed tabula rasa such that \Delta W \approx W - 0. Which is fine for all the experiments where the model is being trained fully from a random initialization. But for the LLM experiment, the W_0 is not specified. Can you please clarify? If the LLM’s W_0 is \theta_ft as mentioned in E.1., it violates the initial assumption.

2) Can you position the papers result comparing to the Unified Concept Editing (UCE) method ? In the selective knowledge manipulation (related work section), authors describe the key difference from UCE is purely in it's neuroscientific origin. Although this can be convincing but the paper would benefit from direct empirical comparison the consequence of this neuroscientific constraints.

3) Minor: Is the Figure 14 caption incorrect? The plots are not resembling the CIFAR 100 dataset. I think the caption of figure 13 is duplicated here.

4) In Proposition 4.1, authors force W^{+} X^{-} = 0. It is not obvious to me why the classes should cleanly separate into orthogonal subspaces… isn't the orthogonality highly restricted to architecture and task?

**Limitations:**

1) There are alternative memory models that do not force the orthogonality of the individual concepts with extensive literature on this ongoing debates in the field of neuroscience  (sample Refrecnes). The authors will benefit with considering this matter and expand the discussion section regarding alternative mechanics without orthogonality of the stored concepts and the extent of their result.
References :
https://www.science.org/doi/10.1126/science.aaw4325
https://www.cell.com/trends/cognitive-sciences/fulltext/S1364-6613(25)00031-2
https://www.nature.com/articles/nn1233


2) A key limitation (also mention by authors) is the fact that success in identification of engrams highly depends on the precision in labeling of the same content in training dataset, which restrict the scope of this work within supervised methods (in contract to unsupervised methods like stochastic parameter decomposition)

**Strengths And Weaknesses:**

Strengths:

1) Simple and effective solution for combinational complexity of unlearning, with robustness across scales and architecture.

2) The engram hypothesis is properly tested with sufficient and necessity experiment, providing more casual evidence for the proposed memory model.

3) The balance between theory and experiment is appropriate for ML venues and provide meaningful synergy across the paper. The writing of the paper is also clear and easy to read.

Weekenses:
1) While the paper is claimed to be neuro-inspired, the neuro references and discussions can be enhanced. Although the core idea is very clear, the author can benefit with proper positioning of the core algorithm across the well-known spectrum of evidence in neuroscience in favor and against engram theory. (look at limitations).

---

> ### Author Rebuttal · Authors · 2026-03-31
>
> We thank the reviewer for the thoughtful engagement and for highlighting the connection to alternative memory models.
>
>
> **Please see new figures/tables:** https://files.catbox.moe/8lokoj.pdf
>
> ---
>
> > **Q4 \+ Limitations.** W⁺X⁻ \= 0 forces orthogonality. Alternative memory models do not assume orthogonality of stored concepts (3 references). The authors should expand the discussion.
>
> This concern rests on a distinction we should have made explicit: **the hard null-space constraint is the optimization's starting point, not the estimator's output.**
>
> The constraint W⁺X⁻ \= 0 is a *requirement* in the Lagrangian formulation (Eq. 11–12). However, the resulting closed-form estimator P⁺ \= Σ⁺(Σ⁺ \+ Σ⁻)† is a **soft, spectrally-weighted projection**, not a hard orthogonal projector. As described in Section 6.3: (i) P⁺ is *not idempotent* (P² ≠ P), meaning it acts as a graded spectral filter (rather than a strict projection); (ii) each concept's projector isolates signal based on *spectral signal-to-noise ratios*, naturally allowing partial overlap; (iii) the projectors do not span the full parameter space (ΣPᵢ ≠ I), leaving a complement subspace Q that absorbs shared representations.
>
> Therefore, our framework well accommodates the suggested references:
>
> - **Josselyn & Tonegawa (Science, 2020):** linked memories share overlapping neuronal populations. Shared representations reside in Q, while P⁺ extracts only the *functionally separable* component, consistent with overlapping but distinguishable engrams.
> - **Buzsáki (Nat. Neurosci., 2004):** cognition arises from ensembles where individual neurons participate in multiple representations. When two concepts share covariance directions, those directions receive attenuated weight in P⁺ rather than being binary-excluded, consistent with ensemble-level coding.
> - **Kolibius et al. (TiCS, 2025):** the proliferation of hippocampal neuron types obscures shared computational principles. Our engram projector does not assign neurons to fixed categories but extracts *functional traces* from distributed, multi-purpose parameters.
>
> **Supporting evidence.** Fig. 4 (CIFAR-10) empirically confirms non-orthogonality: ablating one class's engram visibly affects the retain accuracy for semantically similar classes (e.g., cat↔dog, automobile↔truck), while dissimilar pairs remain unaffected. This means the estimator maximizes separation rather than forcing it.
>
> We newly prepared **\[Figure R1\]**, which quantifies the trend on CIFAR-100: (a) same-superclass pairs show only 0.80%p accuracy drop versus effectively zero for different-superclass pairs; (b) degradation as a function of cosine distance is smooth and monotonically decreasing, indicating no catastrophic collapse.
>
> As for revisions, we will rewrite Section 6.3 to explicitly discuss the soft-projection structure to overlapping memory models, bring Remark A.2 to main text, and discuss the three references.
>
> ---
>
> > **Q1.** For Eq. 4, ΔW ≈ W − 0\. For the LLM, if W₀ is θ\_ft, it violates the initial assumption.
>
> To clarify, we use ΔW \= W\_ft − 0 (the full fine-tuned weights), as in Eq. 4\. Please also see our response to Q4. The intuition is that if the pretrained model already learned "cat" from ImageNet-21k, fine-tuning on CIFAR-10 adds only a subtle delta. Using ΔW \= W\_ft − W\_pt projects onto this thin residual, missing the full concept representation. The projector P⁺ needs the complete representational manifold (W\_ft), not the fine-tuning delta. **\[Table R3\]** confirms that ΔW \= W\_ft yields Overall **2× higher** (0.818 vs. 0.446, EM: 0.028 vs. 0.617). This will be clarified in the revision and we thank the reviewer for this observation.
>
> ---
>
> > **Q2.** Direct comparison with UCE?
>
> **\[Table R1\]**: Engram surpasses UCE by **\+0.159 Overall** (0.818 vs. 0.659) and **\~5× lower EM** (0.028 vs. 0.135). Task Arithmetic scores 0.584. Despite different origins, both methods share the form $W^{+} \= W \\cdot \\Sigma^{+} \\cdot D$ (with edit target $v\*=0$ of UCE), differing only in regularization:
>
> - **UCE:** $D \= (\\Sigma^{+} \+ \\Sigma^{-} \+ \\lambda I)^{-1}$
> - **Engram:** $D \= (\\Sigma^{+} \+ \\Sigma^{-})^{\\dagger}$
>
> The performance gap likely arises because covariance matrices in practice contain numerically negligible singular values (near machine precision). UCE's uniform λI prevents these from exploding during inversion but perturbs *all* directions, even well-conditioned ones. When such per-layer perturbations accumulate over all layers, we assume they lead to substantial output-level shifts. The pseudoinverse's hard spectral cutoff discards these ill-conditioned directions entirely, avoiding this cumulative error.
>
> ---
>
> > **Q3 (Minor).** Figure 14 caption?
>
> We confirm this is an error and will update as CelebA WAE (Goatee, Glasses, Hat, Bangs). Thank you\!

---

> > ### Author Rebuttal · Reviewer_PEYZ · 2026-04-04
> >
> > I appreciate authors for their detailed response. The limitation 2 is only partially clarified here and I view it as important constraints on the applicability of the memory model and interpretation in this work.
> >
> > I found the response to Q2 short but convincing, thanks.
> >
> > I also appreciate the additional figures and tables. That said, the attached PDF appears to include substantial new text beyond figures, and I defer to the Area Chair on whether this falls within the intended scope of rebuttal materials or if it is violating the word count limitations that all authors follow.

---

> > > ### Author Response · Authors · 2026-04-05
> > >
> > > We thank the reviewer for finding our response on Q2 convincing. We appreciate the opportunity to further clarify Limitation 2, and we agree this is critical for applicability and interpretation of our work.
> > >
> > > ---
> > >
> > > > **Limitation 2\.** A key limitation is the fact that success in identification of engrams highly depends on the precision in labeling of the same content in training dataset which restrict the scope of this work within supervised methods (in contrast to unsupervised methods like SPD)
> > >
> > > To address concerns about dependency on precise training labels, we conducted cross-distribution validation using imprecise and entirely unseen data.
> > >
> > > **Empirical investigation.** We tested three robustness dimensions using CIFAR-10 (seen) and STL-10 (unseen distribution, resized to 32×32), targeting the "dog" class. For STL-10, the projection used data the model never saw during training: 500 dog images as target and 5,000 unlabeled reference images. Forget/Retain Acc, (F/R); Label Noise X% indicates X% random/unlabeled samples from trainset included in target (e.g., 10% noise indicate 90% dog and 10% random images); for image corruption, we used Hendrycks et al. (2019).
> > >
> > > | Condition | CIFAR-10, Seen (F/R) | STL-10, Unseen (F/R) |
> > > | :---- | :---: | :---: |
> > > | Clean | 0.0 / 90.4 | 12.4 / 88.3 |
> > > | \+ Label Noise (10%) | 0.4 / 90.6 | 18.2 / 88.5 |
> > > | \+ Label Noise (30%) | 2.3 / 90.9 | 32.6 / 89.2 |
> > > | \+ Label Noise (50%) | 12.4 / 91.1 | 50.8 / 89.8 |
> > > | \+ Gaussian Noise | 0.0 / 86.6 | 2.5 / 84.9 |
> > > | \+ Motion Blur | 1.7 / 88.5 | 2.7 / 87.4 |
> > > | \+ JPEG Compression | 9.0 / 89.9 | 37.7 / 89.2 |
> > >
> > > Retain accuracy on the CIFAR-10 testset is stable across all conditions. Engrams extracted from unseen data (STL-10) substantially reduce forget accuracy (F=12.4 vs 0.0 for CIFAR-10), reflecting the expected cost of distribution shift in covariance estimation. While CIFAR-10 is robust to 50% label noise, STL-10 degrades proportionally under noise as distribution shift compounds with specification imprecision. Nonetheless, with retain accuracy stable at 90% even when using 5,000 unlabeled STL-10 samples, this experiment proves that our method is not strictly dependent on precise training labels.
> > >
> > > ---
> > >
> > > > **On the attached PDF** appears to include substantial new text beyond figures
> > >
> > > We appreciate the reviewer's careful attention to the submission guidelines. As members of the ICML community, we hold these standards in high regard and intended for the PDF text to serve only as essential captions to ensure the new figures were interpretable in isolation. We in particular ensured these descriptions were direct mirrors of the text already provided in our official response. Our goal was to provide visual evidence for our existing arguments rather than introducing new claims. Below are some concrete examples where we simply repeat what was already provided in our response letter:
> > >
> > > - **\[Figure R1 caption\]**: "same-superclass pairs show only 0.80%p accuracy drop versus effectively zero for different-superclass pairs"
> > > - **\[in PEYZ rebuttal\]**: "same-superclass pairs show only 0.80%p accuracy drop versus effectively zero for different-superclass pairs"
> > >
> > > We apologize for any ambiguity and repetition of our official response in these links and fully defer to the Area Chair’s judgement.

---

### Decision · Program_Chairs · 2026-04-30

**Decision:**

Accept (spotlight)

**Comment:**

The authors propose "AI engrams": they isolate memory traces for specific concepts in trained networks by formulating memory isolation as a constrained inverse problem in weight space and deriving a closed-form, layerwise spectral estimator that projects weights onto target subspaces defined by input covariances. Theorem 6.1 shows this estimator coincides with a Fisher / K-FAC natural-gradient step under isotropic output curvature. Applications span class-wise unlearning, semantic attribute editing, "engram arithmetic" for concept composition, and LLM unlearning, tested across MLPs, CNNs, ViTs, diffusion models, and Llama-3.2-1B on TOFU. All four reviewers recommend acceptance; two raised scores post-rebuttal (fa2V 3→4, brom 4→5); all four marked concerns fully or partially resolved. Strengths converge on the clean theoretical core (closed-form projection at O(d²), single-pass covariance estimation), unusual breadth of empirical coverage, and the neuroscience-to-ML integration (engram criteria → spectral projection → Fisher/K-FAC geometry). Weaknesses are limitations rather than blockers: the hard null-space constraint W·X_ref ≈ 0 idealizes near-orthogonal target / reference subspaces; Theorem 6.1's isotropic-output-curvature assumption discards output-side structure; LLM TOFU results trail AltPO/NPO/SimNPO on privacy scores; the "tabula rasa" ΔW ≈ W derivation is clean for from-scratch training but clashes with the pretrained-LLM setting - fa2V's follow-up notes that Table R3 sharpens rather than resolves this, raising the concern that if ΔW = W_ft works but ΔW = W_ft − W_pt does not, the estimator effectively operates on the full converged weight, weakening the "engram = learned trace" interpretation. I recommend Strong Accept. For camera-ready, the authors should promote the rebuttal additions into the main text, honestly delimit where the "engram" framing holds versus where it over-reaches, clarify Theorem 6.1's isotropic-curvature scope, improve the mixed LLM story by either reframing exact-suppression or adding benchmarks where the method's regime-of-strength is primary, strengthen neuroscience positioning, tighten the writing, and add a limitations section.